# ✨ STAR: BOOSTING TIME SERIES FOUNDATION MODELS FOR ANOMALY DETECTION THROUGH STATE-AWARE ADAPTER

## ABSTRACT

While Time Series Foundation Models (TSFMs) have demonstrated remarkable success in Multivariate Time Series Anomaly Detection (MTSAD), in real-world scenarios, many time series comprise not only *numerical variables* such as temperature and flow, but also numerous discrete *state variables* that describe the system status, such as valve on/off or day of the week. Existing TSFMs often overlook the distinct categorical nature of state variables and their critical role as conditions, and typically treat them uniformly with numerical variables. This inappropriate modeling approach prevents the model from fully leveraging state information and even leads to a significant degradation in detection performance after state variables are integrated. To address this critical limitation, this paper proposes a novel **ST**ate-aware **A**dapte**R** (STAR). STAR is a plug-and-play module designed to enhance the capability of TSFMs in modeling and leveraging state variables during the fine-tuning stage. Specifically, STAR comprises three core innovative components: (1) *Identity-guided State Encoder* effectively captures the complex categorical semantics of state variables through a learnable *State Memory*. (2) *Conditional Bottleneck Adapter* dynamically generates low-rank adaptation parameters conditioned on the current state, thereby flexibly injecting the influence of state variables into the backbone model. (3) *Numeral-State Matching* module effectively detects anomalies inherent to the state variables themselves. Extensive experiments conducted on real-world datasets demonstrate that STAR can improve the performance of existing TSFMs on MTSAD.

## 1 INTRODUCTION

In recent years, research into Time Series Foundation Models (TSFMs) has emerged as a significant area of interest. (Shi et al., 2024; Liu et al., 2025; Ekambaram et al., 2024). Researchers are committed to building general-purpose models capable of understanding and processing a diverse range of time series tasks (Liu et al., 2024b; Chen et al., 2025b; Gao et al., 2024). Benefiting from large-scale pre-training, TSFMs can deliver competitive performance without requiring extensive, task-specific fine-tuning. Multivariate Time Series Anomaly Detection (MTSAD), which focuses on identifying abnormal data in multivariate time series, is one of the key downstream tasks targeted by TSFMs (Shentu et al., 2024; Goswami et al., 2024; Wang et al., 2025). It is applied widely in numerous critical domains, including financial fraud detection, medical disease identification, and cybersecurity threat detection (Wen et al., 2022; Yang et al., 2023a; Kieu et al., 2018).

Although TSFMs have achieved impressive performance in MTSAD, real-world industrial scenarios present more complex data manifestations. In these settings, variables exhibit more intricate forms. Beyond common *numerical variables* such as temperature, water flow, and velocity, *state variables* also well exist that describe specific system states, such as the open/closed state of a valve (Mathur & Tippenhauer, 2016), the day of the week (Cui et al., 2016), or the positional state of a machine (von Birgelen & Niggemann, 2018). Unlike their numerical counterparts, these state variables are often organized as discrete data, where each value represents a specific category of that state. Moreover, they provide essential context about the system's operational state and directly influence the numerical variables' magnitude or temporal patterns. We refer to such influence as *condition-based*

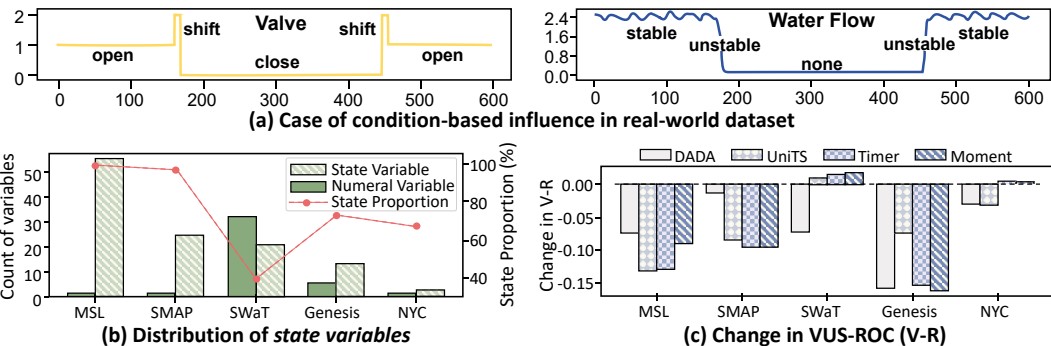

Figure 1: (a) Case of State variables (Valve) and numeral variables (Water Flow) in SWaT. (b) State variables are prevalent in real-world datasets and frequently constitute a substantial proportion. (c) Impact of incorporating state variables on detection performance.

*influence*. For example, as shown in Figure 1a, the valve's state (open, closed, or shifting) is a precondition for water flow and directly dictates the flow pattern (stable, unstable, or none). Figure 1b illustrates that state variables are widely prevalent in common datasets. However, due to either a lack of pre-training on state variables or uniform modeling for both numerical and state variables, **TSFMs often struggle to handle state variables**. As shown in Figure 1c, when these state variables are incorporated into representative TSFMs through unified modeling, the detection performance fails to improve significantly or suffers a substantial decline.

To address this limitation, we aim to develop a plug-and-play method that enables TSFMs to leverage state variables better, thereby enhancing detection performance and facilitating adaptation to a wider range of industrial scenarios. However, this faces the following challenges:

**First, how to extract meaningful embeddings from state variables?** State variables possess complex semantics. For instance, within the SWaT (Mathur & Tippenhauer, 2016), several state variables are primarily grouped into three clusters: switches of valves, pumps, and mercury lamps. Within a given group, state variables share similar semantics and exert similar physical influence on the system. Conversely, state variables across different groups are semantically heterogeneous, manifesting in disparate modes of influence. Furthermore, state variables also exhibit distinct temporal patterns, such as the stable periodic patterns of variables representing the day of the week (Cui et al., 2016). In existing foundation models, state variables are typically embedded using linear layers pre-trained on numerical data. While this approach can partially capture the temporal features, it fails to adequately extract the rich and complex semantics inherent in their states.

**Second, how to model the condition-based influence of state variables?** Different from the correlation among variables (Qiu et al., 2025b; Wu et al., 2025), the state variables act as preconditions that directly determine the magnitude or temporal patterns of the numerical variables. Therefore, for anomaly detection, it is essential to consider the condition-based influence when processing numerical variables. However, this asymmetric influence between state and numerical variables renders the modeling of condition-based influence a non-trivial task. Some TSFMs, such as UniTS (Gao et al., 2024), model interactions among all variables in a unified manner, this approach erroneously treats them as equivalent, thereby failing to model the condition-based influence effectively.

To close these gaps, we propose **STAR**, a novel **ST**ate-aware **A**dapte**R** for TSFMs in the MTSAD to better model state variables during the fine-tuning phase. **First**, to better extract embeddings from state variables, we designed a *State Extractor* capable of simultaneously capturing both the state information and temporal patterns of state variables. This module incorporates an *Identity-guided State Encoder* which establishes a small-scale State Memory and retrieves a composite memory's representation to encode state information guided by both Variable Identity and State Identity. This method effectively models the complex semantics of state variables. **Second**, we propose a *Conditional Bottleneck Adapter* to effectively model the condition-based influence. The adapter dynamically generates adaptation parameters and bottleneck size based on state variables to modulate the TSFM's parameters, thereby directly influencing how the TSFMs process numerical variables. Building upon the aforementioned modules, we can already capture meaningful state information and process numerical variables based on state. Furthermore, we have developed a *Numeral-State Matching* module to more effectively detect anomalies contained within the state variables.

Our contributions are summarized as follows:

- We design a universal State-aware Adapter that equips TSFMs with a more effective mechanism for handling state variables for the MTSAD.

- We propose a novel *Identity-guided State Encoder* in STAR to effectively model the complex homogeneous and heterogeneous semantics of state variables.

- We propose a novel *Conditional Bottleneck Adapter* in STAR, which can flexibly incorporate the condition-based influence of state variables in the TSFMs.

- We conducted extensive experiments on real-world datasets. The results show that STAR effectively improves the performance of TSFMs in MTSAD.

## 2 RELATED WORKS

### 2.1 MULTIVARIATE TIME SERIES ANOMALY DETECTION

Classic methods for MTSAD can be classified into non-learning (Breunig et al., 2000; Li et al., 2024b), machine learning (Liu et al., 2008; Ramaswamy et al., 2000), and deep learning (Wu et al., 2025; Zhong et al., 2025b; Liu et al., 2024a; Liu & Paparrizos, 2024). In particular, methods based on deep learning have seen significant success in recent years. They can be classified into forecasting-based (Deng & Hooi, 2021), reconstruction-based (Wu et al., 2023; Luo & Wang, 2024; Tuli et al., 2022) and contrastive-based methods (Xu et al., 2021; Yang et al., 2023b; Guo et al., 2023). Although some of these models consider variable interactions within a unified modeling framework (Wu et al., 2025; Xie et al., 2025), or employ specialized modeling for discrete variables (Li et al., 2024a; Chen et al., 2025a), they still treat state variables as mere discrete numerical values, thereby overlooking the condition-based influence of state variables.

### 2.2 TIME SERIES FOUNDATION MODELS FOR ANOMALY DETECTION

Researchers are actively exploring Foundation Models for various time series analysis tasks, including forecasting (Wang et al., 2025; Liu et al., 2025; Wang et al., 2024; Das et al., 2024), classification (Chen et al., 2025b), and anomaly detection (Shentu et al., 2024). For anomaly detection, models can be classified into two primary categories: 1) **Task-general model**: These models are typically pre-trained on extensive data, endowing them with excellent representation learning capabilities (Gao et al., 2024; Goswami et al., 2024; Liu et al., 2024b). 2) **Task-specific model**: These models have architectures specifically designed for the anomaly detection task. For instance, to better adapt to anomalous data, DADA (Shentu et al., 2024) employs multiple bottleneck layers of varying sizes for decoding. Additionally, some work has focused on designing adapters to augment TSFMs during fine-tuning phase, such as AdaPTS (Benechehab et al., 2025) and MSFT (Qiao et al., 2025). However, these efforts are primarily concentrated on forecasting tasks, and more importantly, fail to enable TSFMs to better process state variables.

## 3 STAR

For the task of time series anomaly detection, we consider a time series as $\boldsymbol{X} = \{\boldsymbol{X}^n, \boldsymbol{X}^s\} \in \mathbb{R}^{T \times (C_n + C_s)}$, which consists of $C_n$ numerical variables $\boldsymbol{X}^n$ and $C_s$ state variables $\boldsymbol{X}^s$ over $T$ time points. A TSFM is leveraged to identify anomalies in a previously unseen test series $\boldsymbol{X}_{\text{test}}$ either via direct inference or subsequent fine-tuning. Specifically, the model's goal is to generate a binary prediction sequence $\boldsymbol{Y}_{\text{test}} = (y_1, y_2, \ldots, y_{T_{\text{test}}}) \in \mathbb{R}^{T_{\text{test}}}$. Each element $y_t \in \{0, 1\}$ in this sequence serves as a label, signifying whether the corresponding data point $y_t$ at time $t$ is anomalous.

### 3.1 OVERVIEW

Figure 2 shows the overall architecture of **STAR**, which reconsiders the state variables for TSFMs using the **ST**ate-aware **A**dapte**R**. We shift the processing of state variables from TSFMs to STAR. It mainly consists of three modules: 1) **State Extractor**. Addressing the specific characteristics of state variables, this module simultaneously extracts their state and temporal features and utilizes an

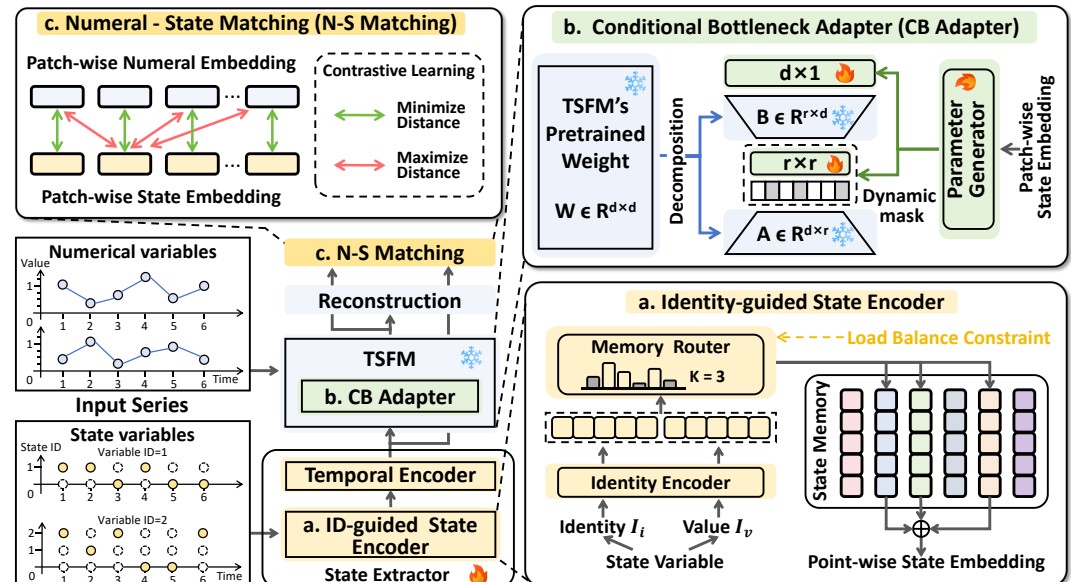

Figure 2: The overview of STAR.

*Identity-guided Selective State Encoder* to model the complex semantics of state variables effectively. 2) **Conditional Bottleneck Adapter**. To better model the condition-based influence, we no longer directly model interactions between state and numerical variables. Instead, we use the state variable to modulate the TSFM's parameters, thereby influencing its processing of numerical variable. 3) **Numeral-State Matching**. To better detect anomalies in the state, we design a contrastive learning based on the degree of matching between numerical and state variables at the patch level. Notably, during the fine-tuning process, the entire Backbone network is frozen, and only STAR is utilized to perform the fine-tuning effectively.

## 3.2 State Extractor Guided by Identity

The *State Extractor* operates in two sequential stages. First, an *Identity-guided State Encoder* processes each state variable to capture its information from both the state identity (i.e., index of the state variable) and the state value (i.e., discrete value of the state variable) at each time point. Subsequently, a *Temporal Encoder* extracts temporal patterns within each patch.

Specifically, the state identity serves as the unique, static identifier for a state variable within the multivariate time series, corresponding directly to its channel index. For example, a variable located at index $i$ has a variable identity of $i$.

In contrast, the state value is the discrete value of a state variable at a specific time point. For instance, if a state variable has a value of $j$ at a given time step, its state value is $j$. Note that these values are mapped to non-negative integers (e.g., {0, 1, 2, ...}) during data preprocessing.

### 3.2.1 Identity-guided State Encoder

Formally, state variables are a type of categorical variable, and different state variables can have different cardinality or the number of categories. Most existing methods learn a representation for every individual category via a dedicated, learnable vector to embed the categorical variable (Shan et al., 2016; Hollmann et al., 2025; Li et al., 2025). This method, however, imposes a substantial parameter burden and learning complexity. More importantly, it fails to account for the similarity or heterogeneity among different state variables.

To address the complex semantics of state variables associated with grouping patterns, we propose a novel router network that dynamically assigns different state variables that have distinct state values into clusters. Our approach introduces a learnable, compact State Memory, denoted as $\boldsymbol{S} \in \mathbb{R}^{N \times d}$, which consists of $N$ $d$-dimensional vectors corresponding to distinct groups. Under the guidance of

the embedding of state identity $\boldsymbol{I}_i \in \mathbb{R}^{T \times C_s \times d}$ and the embedding of state value $\boldsymbol{I}_v \in \mathbb{R}^{T \times C_s \times d}$, we select a vector subset from the State Memory and utilize the linear combination of its vectors to represent the state variable. We begin by employing a Sinusoidal Encoding (Vaswani et al., 2017) to transform the state identity and state value into $\boldsymbol{I}_i$ and $\boldsymbol{I}_v$:

$$\boldsymbol{I}_i[:,i,k] = \begin{cases} \sin\left(\frac{i}{\lambda^{2n/d}}\right), & \text{if } j = 2n \\ \cos\left(\frac{i}{\lambda^{2n/d}}\right), & \text{if } j = 2n+1 \end{cases}, \quad \boldsymbol{I}_v[:,i,k] = \begin{cases} \sin\left(\frac{\boldsymbol{X}^S[:,i]}{\lambda^{2n/d}}\right), & \text{if } j = 2n \\ \cos\left(\frac{\boldsymbol{X}^S[:,i]}{\lambda^{2n/d}}\right), & \text{if } j = 2n+1 \end{cases}, \quad (1)$$

where $\lambda$ is a constant; $i$ and $k$ represent the index of the variable and the hidden dimension of the embedding, respectively; $n \in \{0, 1, 2, 3 \dots\}$ represents any natural number with $2n$ and $2n+1$ corresponding to even and odd components; and $\boldsymbol{X}^S[:,i]$ denote the discrete value of the $i$-th state variable at the corresponding time point. The *Memory Router* leverages $\boldsymbol{I}_i$ and $\boldsymbol{I}_v$ to dynamically choose a subset of K vectors from the State Memory. To preserve a continuous gradient flow, we implement the selection mechanism using a soft mask:

$$\boldsymbol{W}_s = f(\text{concat}(\boldsymbol{I}_i, \boldsymbol{I}_v)) \in \mathbb{R}^{T \times C_s \times N}, \quad \boldsymbol{\theta} = \text{topK}(\boldsymbol{W}_s) \in \mathbb{R}^{T \times C_s \times 1}, \quad (2)$$

$$\boldsymbol{W}_{\text{mask}} = \text{softmax}(\boldsymbol{W}_s + \log(\text{sigmoid}((\boldsymbol{W}_s - \boldsymbol{\theta})/\epsilon))) \in \mathbb{R}^{T \times C_s \times N}, \quad (3)$$

where $f$, implemented as an MLP, learns the selection weights from the concatenated embeddings. The function topK returns the K-th largest value, denoted as $\theta$. $\boldsymbol{W}_{\text{mask}}$ represents the weights after applying the soft mask, where all values smaller than $\boldsymbol{\theta}$ are attenuated towards zero. $\epsilon$ is a small positive constant that pushes the sigmoid's output to its extremes (0 or 1). The point-wise state embedding $\boldsymbol{S}_{\text{point}} \in \mathbb{R}^{T \times C_s \times d}$ is obtained via the equation: $\boldsymbol{S}_{\text{point}} = \boldsymbol{W}_{\text{mask}} \otimes \boldsymbol{S}$.

To further promote the comprehensive training of all elements within the State Memory and to prevent routing imbalance, we introduce a constraint based on the Coefficient of Variation (Lovie, 2005). This constraint is applied to both the selection frequency $\boldsymbol{E}_{\text{sel}}$ and the routing importance of each element $\boldsymbol{E}_{\text{imp}}$ to ensure load balancing:

$$\boldsymbol{E}_{\text{sel}} = \text{avg}_{T,C_s}(\text{sigmoid}((\boldsymbol{W}_s - \boldsymbol{\theta})/\epsilon)) \in \mathbb{R}^N, \quad \boldsymbol{E}_{\text{imp}} = \text{avg}_{T,C_s}(\boldsymbol{W}_{\text{mask}}) \in \mathbb{R}^N, \quad (4)$$

$$\mathcal{L}_{\text{bal}} = \left(\frac{\text{var}_N(\boldsymbol{E}_{\text{sel}})}{\text{avg}_N(\boldsymbol{E}_{\text{sel}})}\right)^2 + \left(\frac{\text{var}_N(\boldsymbol{E}_{\text{imp}})}{\text{avg}_N(\boldsymbol{E}_{\text{imp}})}\right)^2, \quad (5)$$

where $\text{avg}_{T,C_s}(\cdot)$ denotes average aggregation along the temporal and variable dimensions, $\text{avg}_N(\cdot)$ and $\text{var}_N(\cdot)$ are functions that compute the mean and variance of $\boldsymbol{E}_{\text{sel}}$ and $\boldsymbol{E}_{\text{imp}}$, respectively. $\mathcal{L}_{\text{bal}}$ denotes the load-balancing loss, which is incorporated into the overall training objective.

### 3.2.2 TEMPORAL ENCODER

Motivated by the observation that state variables often exhibit temporal patterns, such as the strong periodicity representing the day of the week, we further model the temporal interactions among the point-wise state embedding $\boldsymbol{S}_{\text{point}}$ to derive patch-wise state embeddings $\boldsymbol{S}_{\text{patch}}$. Initially, we segment $\boldsymbol{S}_{\text{point}}$ into $m$ patches of length $l$ in a manner consistent with the backbone's configuration, yielding $\boldsymbol{P} \in \mathbb{R}^{m \times l \times C_s \times d}$. This process comprises two components: intra-patch and inter-patch interaction:

$$\boldsymbol{P}_{\text{intra}} = f_1(\boldsymbol{P}) \in \mathbb{R}^{m \times C_s \times d}, \quad \boldsymbol{P}_{\text{inter}} = f_2(\boldsymbol{P}_{\text{intra}}) + \boldsymbol{P}_{\text{intra}} \in \mathbb{R}^{m \times C_s \times d}, \quad (6)$$

where $f_1 : \mathbb{R}^{l \times d} \to \mathbb{R}^d$ aggregates the representations within each patch, and $f_2 : \mathbb{R}^m \to \mathbb{R}^m$ extracts valuable contextual information for each patch. Each state variable reflects only a partial aspect of the system's information. Therefore, to construct a more holistic representation of the system's overall state, we further perform an aggregation over these variables inside each patch:

$$\boldsymbol{W}_{\text{agg}} = f_3(\boldsymbol{P}_{\text{inter}}) \in \mathbb{R}^{m \times C_s}, \quad \boldsymbol{S}_{\text{patch}} = f_4(\text{sum}_{C_s}(\boldsymbol{W}_{\text{agg}} \odot \boldsymbol{P}_{\text{inter}})) \in \mathbb{R}^{m \times d}, \quad (7)$$

where $f_3 : \mathbb{R}^d \to \mathbb{R}^1$ is a function that adaptively computes the aggregation weights, and $f_4 : \mathbb{R}^d \to \mathbb{R}^d$ serves to refine the resulting fused representation further, $\odot$ denotes element-wise multiplication. For a lightweight implementation, both $f_1$, $f_2$, $f_3$, and $f_4$ are implemented as MLPs.

## 3.3 CONDITIONAL BOTTLENECK ADAPTER

To efficiently fine-tune pre-trained parameters, the weight update $\Delta \boldsymbol{W}$ for a pre-trained matrix $\boldsymbol{W}_0 \in \mathbb{R}^{d_{\text{in}} \times d_{\text{out}}}$ is often constrained to a low-rank decomposition (Hu et al., 2022). This process can be formalized as:

$$\boldsymbol{h} = \boldsymbol{W}_0 \boldsymbol{x} + \Delta \boldsymbol{W} \boldsymbol{x}, \Delta \boldsymbol{W} = \boldsymbol{A} \boldsymbol{B}, \tag{8}$$

where $\boldsymbol{h}$ and $\boldsymbol{x}$ denote the input and output of the module, respectively. $\boldsymbol{A} \in \mathbb{R}^{d_{\text{in}} \times r}$ and $\boldsymbol{B} \in \mathbb{R}^{r \times d_{\text{out}}}$ are two trainable low-rank matrices, with the rank $r \ll \min(d_{\text{in}}, d_{\text{out}})$.

Leveraging the efficiency of low-rank decomposition, we innovatively propose the Conditional Bottleneck Adapter. This module perceives the system's state to dynamically generate its parameters and determine its bottleneck size, as illustrated in Figure 2b. The weight update $\Delta \boldsymbol{W}$ is given by the following equation:

$$\Delta \boldsymbol{W} = \boldsymbol{A} \boldsymbol{R} \boldsymbol{B} \odot \boldsymbol{D}, \ \ \boldsymbol{R} \in \mathbb{R}^{r \times r}, \boldsymbol{D} \in \mathbb{R}^{d_{\text{out}} \times 1}, \tag{9}$$

where $\boldsymbol{A}$ and $\boldsymbol{B}$ are decomposed from $\boldsymbol{W}_0$, as detailed in Section 3.3.1, and remain frozen. $\boldsymbol{R}, \boldsymbol{D}$ are determined by the state variable as detailed in Section 3.3.2.

Different from selecting the bottleneck size with the Mixture of Experts architecture (Shentu et al., 2024), which has larger parameters, we propose a novel dynamic mask on the $\boldsymbol{R}$ matrix to modulate the bottleneck size efficiently. This mechanism enables the model to adaptively filter information based on the characteristics of diverse system states. With the low-rank matrix decomposition and the learnable matrix $\boldsymbol{R}$ condition on the system's state, our method avoids additional performance overhead. Theoretically, our adapter can be integrated with the linear layers of any module.

### 3.3.1 PRETRAINED WEIGHT DECOMPOSITION

To preserve the performance of the original parameters as much as possible, we employ Singular Value Decomposition (SVD) (Stewart, 1993) to decompose the pretrained weight:

$$\boldsymbol{U}, \boldsymbol{\Sigma}, \boldsymbol{V} = \text{SVD}(\boldsymbol{W}_0), \boldsymbol{U} \in \mathbb{R}^{d_{\text{in}} \times d_{\text{in}}}, \boldsymbol{\Sigma} \in \mathbb{R}^{d_{\text{in}} \times d_{\text{out}}}, \boldsymbol{V} \in \mathbb{R}^{d_{\text{out}} \times d_{\text{out}}}, \tag{10}$$

where $\boldsymbol{\Sigma}$ is the singular value matrix, which we truncate to its leading $r$ values to form the matrix $\boldsymbol{\Sigma}_1 \in \mathbb{R}^{d_{\text{in}} \times r}$ and $\boldsymbol{\Sigma}_2 \in \mathbb{R}^{r \times d_{\text{out}}}$ for the purpose of low-rank approximation. Then we compute the low-rank matrices as :

$$\boldsymbol{A} = \boldsymbol{U} \boldsymbol{\Sigma}_1 \in \mathbb{R}^{d_{\text{in}} \times r}, \boldsymbol{B} = \boldsymbol{\Sigma}_2 \boldsymbol{V} \in \mathbb{R}^{r \times d_{\text{out}}}. \tag{11}$$

### 3.3.2 PARAMETER GENERATOR AND DYNAMIC MASK

For the $t$-th patch, we utilize its corresponding $\boldsymbol{S}_{\text{patch}}^t$ to compute the state-dependent parameters $\boldsymbol{R}$ and $\boldsymbol{D}$. It is worth noting that the $\boldsymbol{R}$ and $\boldsymbol{D}$ matrices are patch-specific, as they are dynamically generated based on the distinct state of each individual patch:

$$\boldsymbol{R}_{\text{init}}, \boldsymbol{D} = g_1(\boldsymbol{S}_{\text{patch}}^t), g_1 := \mathbb{R}^d \to \mathbb{R}^{(r \times r + d_{\text{out}})}. \tag{12}$$

As shown in Figure 2b, we adaptively learn a state-aware dynamic mask matrix based on state variables via a soft-masking mechanism. We apply this mask to the vector $\boldsymbol{R}_{init}$ and then insert the masked vector between the low-rank matrices $\boldsymbol{A}$ and $\boldsymbol{B}$ to modulate the bottleneck size of the Conditional Bottleneck (CB) Adapter. This process can be formalized as follows:

$$\boldsymbol{R}_{\text{mask}} = g_2(\boldsymbol{S}_{\text{patch}}^t) \in \mathbb{R}^r, \Gamma = g_3(\boldsymbol{S}_{\text{patch}}^t), \tag{13}$$

$$\boldsymbol{M} = \text{sigmoid}((\boldsymbol{R}_{\text{mask}} - \Gamma)/\epsilon) \in \mathbb{R}^{r \times 1}, \boldsymbol{R} = \boldsymbol{M} \odot \boldsymbol{R}_{\text{init}}, \tag{14}$$

where $g_2 := \mathbb{R}^d \to \mathbb{R}^r$, $g_3 := \mathbb{R}^d \to \mathbb{R}^1$. $g_1, g_2, g_3$ are implemented as MLPs. $\boldsymbol{R}_{\text{mask}}$ is score for each row in $\boldsymbol{R}$, $\Gamma$ is a masking threshold. $\boldsymbol{M}$ serves as the soft mask matrix, where if an element in $\boldsymbol{R}_{\text{mask}}$ is below the threshold $\Gamma$, the corresponding elements in $\boldsymbol{M}$ are driven towards zero. Leveraging a dynamic mask mechanism, the $\boldsymbol{R}$ matrix adaptively controls the bottleneck size according to the system state, allowing for the dynamic handling of numerical variables.

### 3.4 PIPELINE WITH NUMERAL - STATE MATCHING

Anomalies may manifest not only as aberrant temporal patterns in numerical variables but also as abnormal conditions within the state variables. Existing TSFMs, however, detect anomalies solely through time series reconstruction. Since state variables are represented as categorical variables, merely reconstructing their state identities is insufficient for accurately identifying state-based anomalies. To address this limitation, we introduce the *Numeral-State Matching* module. This module is designed to detect anomalies by learning the degree of correspondence between numerical variables and their associated state variables.

#### 3.4.1 FINETUNING PHASE.

Constructing positive and negative pairs for contrastive learning by injecting anomalies introduces significant training overhead and makes it difficult to guarantee the effectiveness of the synthetic anomalies (Zhong et al., 2025a; Kim et al., 2025). To circumvent these issues, we adopt a different approach. As shown in Figure 2c, we define a positive pair as the corresponding patch-wise numeral embedding $\boldsymbol{N}_{\text{patch}}^t \in \mathbb{R}^d$ and patch-wise state embedding $\boldsymbol{S}_{\text{patch}}^t \in \mathbb{R}^d$. All other non-corresponding pairings are consequently treated as negative examples. Instead of contrasting among different views of the same variable (Kim et al., 2024; Fang et al., 2024), our method focuses on learning the correlations between state and numerical variables, and subsequently uses this as a basis for detecting potential anomalies. The loss for the Numeral - State Matching Contrastive Learning can be expressed as follows:

$$\mathcal{L}_{\text{match}} = -\frac{1}{M} \sum_{t=1}^{M} \log\left(\frac{\exp(\text{sim}(\boldsymbol{N}_{\text{patch}}^t, \boldsymbol{S}_{\text{patch}}^t)/\tau)}{\sum_{k=1}^{M} \exp(\text{sim}(\boldsymbol{N}_{\text{patch}}^k, \boldsymbol{S}_{\text{patch}}^t)/\tau))}\right), \quad (15)$$

where $M$ denotes the total number of patches in one batch. $\boldsymbol{N}_{\text{patch}}^i$ and $\boldsymbol{S}_{\text{patch}}^i$ are the respective numeral and state embeddings corresponding to the $i$-th patch. The function $\text{sim}(\cdot)$ is the cosine similarity, and $\tau$ is a temperature coefficient used to scale the logits in the contrastive loss.

The overall loss during training is composed of three components: the numerical variable reconstruction loss ($\mathcal{L}_{\text{rec}}$), the load-balancing loss ($\mathcal{L}_{\text{balance}}$), and the Numeral-State Matching loss ($\mathcal{L}_{\text{match}}$). The total loss, $\mathcal{L}_{\text{total}}$, is formulated as: $\mathcal{L}_{\text{total}} = \mathcal{L}_{\text{rec}} + \lambda_1 \mathcal{L}_{\text{balance}} + \lambda_2 \mathcal{L}_{\text{match}}$. where $\lambda_1$ and $\lambda_2$ are hyperparameters that balance the contribution of each loss term.

#### 3.4.2 INFERENCE PHASE

Anomaly detection is performed in two parts: point-wise *numerical reconstruction* and patch-wise *Numeral-State Matching*. During inference on a test time series, their respective scores, denoted as $\text{score}_{\text{rec}}$ and $\text{score}_{\text{match}}$, are calculated separately:

$$\text{score}_{rec} = \text{avg}_{C_N}(\text{mse}(\hat{\boldsymbol{X}}_{\text{test}}^n, \boldsymbol{X}_{\text{test}}^n)) \in \mathbb{R}^{T_{\text{test}}}, \ \ \text{score}_{match} = \text{sim}(\boldsymbol{N}_{\text{patch}}, \boldsymbol{S}_{\text{patch}}) \in \mathbb{R}^{M_{\text{test}}}, \quad (16)$$

where $\text{mse}(\cdot)$ is the Mean Square Error, $\text{sim}(\cdot)$ denotes the cosine similarity, $\hat{\boldsymbol{X}}^n$ represents the reconstruction output of numerical variables from the backbone, and $T_{\text{test}}$ and $m_{\text{test}}$ are the total number of time points and patches in the test set, respectively. $\text{avg}_{C_N}(\cdot)$ is an operation that computes the mean value across the variable dimension.

Given that $\text{score}_{\text{rec}}$ and $\text{score}_{\text{match}}$ differ in both temporal granularity and numerical scale, we first apply linear interpolation to $\text{score}_{\text{match}}$ to align their resolutions. Subsequently, the two scores are fused by taking their element-wise product:

$$\text{score}_{\text{match}}' = \text{Linear-Interpolation}(\text{score}_{\text{match}}) \in \mathbb{R}^{T_{\text{test}}}, \quad (17)$$

$$\text{score}_{\text{total}} = \text{score}_{rec} \times \text{softmax}(\text{score}_{\text{match}}') \in \mathbb{R}^{T_{\text{test}}}. \quad (18)$$

Our approach not only detects anomalies in numerical variables but also perceives anomalous in the system state. Furthermore, by leveraging patch-wise similarity computation, the model's capability to detect subsequence anomalies is effectively enhanced.

# 4 EXPERIMENTS

## 4.1 EXPERIMENTAL SETTINGS

**Datasets** We evaluate STAR on various datasets with state variables, including **MSL** (Mars Science Laboratory Dataset) (Hundman et al., 2018), **SMAP** (Soil Moisture Active Passive Dataset) (Hundman et al., 2018), **SWaT** (Secure Water Treatment) (Mathur & Tippenhauer, 2016), **Genesis** (von Birgelen & Niggemann, 2018), and **NYC** (Cui et al., 2016). The statistical details about the datasets are available in Appendix A.1.

**Backbone** We choose the latest state-of-the-art TSFMs for anomaly detection as backbones, including DADA (Shentu et al., 2024), UniTS (Gao et al., 2024), Moment (Goswami et al., 2024), and Timer (Liu et al., 2024b). The detailed description is available in Appendix A.3.

**Metrics** To circumvent the potential for unfair comparisons arising from the diverse threshold selection methods employed by different TSFMs, we primarily adopt threshold-irrelevant metrics for evaluation. These include the AUC-ROC (A-R), VUS-ROC (V-R), and VUS-PR (V-P) (Paparrizos et al., 2022). Meanwhile, the comprehensive evaluation is also supplemented by a range of commonly used metrics mentioned in TAB(Qiu et al., 2025a). The description and implementation details of all metrics are shown in Appendix A.4.

## 4.2 DETECTION RESULTS

**Main results** We evaluate our model on five real-world datasets containing state variables, using four TSFMs as backbones. We uniformly use 10% of the downstream data for fine-tuning. The results are summarized in Table 1. For a fair comparison, we evaluated two baseline fine-tuning methods: (1) **Backbone**: standard fine-tuning, and (2) **LoRA**: fine-tuning the base model with LoRA (Hu et al., 2022). It can be seen that LoRA-based fine-tuning offers inconsistent and often marginal performance changes, our method consistently delivers substantial enhancements to the detection performance of TSFMs. Compared to standard fine-tuning, STAR improves the V-R metric on task-specific and task-general TSFMs by 6.93% and 6.07%, respectively. This demonstrates the generalizability of STAR across different types of TSFMs. The validity of our motivation and modeling is further highlighted on the Genesis and NYC, where STAR delivers more improvements of 8.73% and 9.17%, owing to the meaningful physical interpretations of their state variables. Notably, despite the significant performance discrepancies on the SWaT dataset, this variance is considered to be within a reasonable range due to the unique characteristics of its distribution. A similar phenomenon can be found in Wu et al. (2025).

**Multi-metrics** To ensure a comprehensive comparison, we evaluate the models on more metrics, including Accuracy (ACC), Range-Precision (R-P), Range-Recall (R-R), Range-F1-score (R-F1) (Tatbul et al., 2018), Affiliated-Precision (Aff-P), Affiliated-Recall (Aff-R), Affiliated-F1 (Aff-f1) (Huet et al., 2022), AUC-PR (A-P), R-AUC-PR (R-A-P), R-AUC-ROC (R-A-R) (Paparrizos et al., 2022). Table 2 shows that STAR yields performance improvements across the majority of metrics.

## 4.3 MODEL ANALYSIS

**Ablation Study** To analyze the contribution of each module, we conducted an ablation study, detailed in Table 3. Row 1 shows the standard fine-tuning, while Rows 2-6 illustrate the performance when integrating specific modules of STAR with the backbone. We have the following observations: (1) A comparison of rows 1-3 reveals that both the *Conditional Bottleneck Adapter* (*CB Adapter*) and the *Numeral-State Matching* (*N-S Matching*) module can enhance the backbone's performance to a certain degree. This limited improvement can likely be attributed to the lack of meaningful state information. (2) In Rows 4 and 5, the addition of the *ID-guided State Encoder* results in a more pronounced performance enhancement with the two modules. This validates the effectiveness of the *ID-guided State Encoder* in capturing complex state information. (3) The combination of all three modules creates a virtuous cycle, resulting in the best performance, as shown in Row 6. Specifically, the *N-S Matching* module optimizes the state representations, which in turn enhances the effectiveness of the other modules.

Table 1: Results of adapters for different TFSMs. The better results are highlighted in **bold**.

| Dataset | MSL | | | SMAP | | | SWaT | | | Genesis | | | NYC | | |
|---|---|---|---|---|---|---|---|---|---|---|---|---|---|---|---|
| Metric | A-R | V-R | V-P | A-R | V-R | V-P | A-R | V-R | V-P | A-R | V-R | V-P | A-R | V-R | V-P |
| DADA | 0.702 | 0.746 | 0.284 | 0.406 | 0.451 | 0.123 | 0.814 | 0.740 | 0.558 | 0.802 | 0.832 | 0.154 | 0.492 | 0.652 | 0.050 |
| +LoRA | 0.721 | 0.765 | 0.298 | 0.420 | 0.463 | 0.129 | 0.807 | 0.735 | 0.562 | 0.784 | 0.818 | 0.153 | 0.514 | 0.666 | 0.052 |
| +STAR | **0.787** | **0.803** | **0.323** | **0.447** | **0.490** | **0.129** | **0.838** | **0.753** | **0.577** | **0.896** | **0.911** | **0.165** | **0.571** | **0.701** | **0.065** |
| UniTS | 0.747 | 0.794 | 0.296 | 0.515 | 0.555 | 0.145 | 0.236 | 0.337 | 0.126 | 0.704 | 0.770 | 0.017 | 0.486 | 0.613 | 0.044 |
| +LoRA | 0.748 | 0.796 | 0.298 | 0.529 | 0.567 | 0.148 | 0.235 | 0.336 | 0.125 | 0.706 | 0.777 | 0.018 | 0.508 | 0.622 | 0.049 |
| +STAR | **0.776** | **0.810** | **0.332** | **0.535** | **0.575** | **0.154** | **0.242** | **0.355** | **0.137** | **0.809** | **0.850** | **0.038** | **0.595** | **0.717** | **0.082** |
| Moment | 0.714 | 0.753 | 0.281 | 0.520 | 0.561 | 0.145 | 0.267 | 0.376 | 0.185 | 0.886 | 0.906 | 0.155 | 0.521 | 0.632 | 0.045 |
| +LoRA | 0.716 | 0.755 | 0.287 | 0.521 | 0.563 | 0.145 | 0.272 | 0.383 | 0.190 | 0.889 | 0.908 | 0.184 | 0.526 | 0.636 | 0.045 |
| +STAR | **0.771** | **0.807** | **0.309** | **0.524** | **0.562** | **0.146** | **0.283** | **0.388** | **0.198** | **0.949** | **0.950** | **0.254** | **0.576** | **0.680** | **0.050** |
| Timer | 0.749 | 0.793 | 0.299 | 0.530 | 0.571 | 0.149 | 0.226 | 0.326 | 0.125 | 0.842 | 0.854 | 0.100 | 0.517 | 0.628 | 0.043 |
| +LoRA | 0.750 | 0.789 | 0.311 | 0.535 | 0.575 | 0.155 | 0.234 | 0.337 | 0.126 | 0.902 | 0.887 | 0.158 | 0.455 | 0.602 | 0.039 |
| +STAR | **0.786** | **0.812** | **0.344** | **0.538** | **0.579** | **0.154** | **0.244** | **0.352** | **0.129** | **0.931** | **0.944** | **0.179** | **0.541** | **0.658** | **0.050** |

Table 2: Multi-metrics results on real-world datasets. The better results are highlighted in **bold**.

| Dataset | Method | Threshold-relevant | | | | | | | Threshold-irrelevant | | | | | |
|---|---|---|---|---|---|---|---|---|---|---|---|---|---|---|
| | | ACC | R-P | R-R | R-F1 | Aff-P | Aff-R | Aff-F1 | A-R | A-P | R-A-R | R-A-P | V-R | V-P |
| MSL | DADA | 0.814 | **0.179** | 0.288 | 0.221 | 0.603 | **0.981** | 0.747 | 0.702 | 0.232 | 0.752 | 0.290 | 0.746 | 0.284 |
| | +STAR | **0.822** | 0.164 | **0.346** | **0.223** | **0.694** | 0.952 | **0.803** | **0.790** | **0.276** | **0.811** | **0.337** | **0.807** | **0.331** |
| | UniTS | 0.747 | **0.197** | 0.261 | 0.224 | 0.649 | **0.996** | 0.786 | 0.747 | 0.222 | 0.802 | 0.303 | 0.794 | 0.296 |
| | +STAR | **0.816** | 0.185 | **0.337** | **0.239** | **0.692** | 0.955 | **0.803** | **0.776** | **0.255** | **0.815** | **0.341** | **0.810** | **0.332** |
| Genesis | DADA | 0.987 | 0.333 | 0.162 | 0.218 | 0.835 | 0.953 | 0.890 | 0.802 | **0.175** | 0.844 | 0.152 | 0.832 | 0.154 |
| | +STAR | **0.996** | **0.571** | **0.172** | **0.264** | **0.895** | **0.969** | **0.930** | **0.896** | 0.159 | **0.921** | **0.156** | **0.911** | **0.165** |
| | UniTS | 0.977 | 0.031 | **0.165** | 0.058 | 0.704 | **0.976** | 0.818 | 0.704 | 0.017 | 0.791 | 0.018 | 0.770 | 0.017 |
| | +STAR | **0.995** | **0.154** | 0.156 | **0.155** | **0.807** | 0.949 | **0.872** | **0.809** | **0.047** | **0.861** | **0.039** | **0.850** | **0.038** |

Table 3: Ablation studies for STAR. The better results are highlighted in **bold**.

| | Dataset | | | MSL | | | | Genesis | | | |
|---|---|---|---|---|---|---|---|---|---|---|---|
| ID-guided State Encoder | CB Adapter | N - S Matching | | DADA | | UniTS | | DADA | | UniTS | |
| | | | | V-R | V-P | V-R | V-P | V-R | V-P | V-R | V-P |
| 1 | ✗ | ✗ | ✗ | 0.746 | 0.284 | 0.794 | 0.296 | 0.832 | 0.154 | 0.770 | 0.017 |
| 2 | ✗ | ✓ | ✗ | 0.762 | 0.295 | 0.792 | 0.299 | 0.855 | 0.154 | 0.784 | 0.022 |
| 3 | ✗ | ✗ | ✓ | 0.755 | 0.290 | 0.800 | 0.298 | 0.841 | 0.157 | 0.780 | 0.019 |
| 4 | ✓ | ✓ | ✗ | 0.803 | 0.323 | 0.807 | 0.311 | 0.872 | 0.156 | 0.819 | 0.021 |
| 5 | ✓ | ✗ | ✓ | 0.760 | 0.295 | 0.805 | 0.321 | 0.854 | 0.150 | 0.805 | 0.024 |
| 6 | ✓ | ✓ | ✓ | **0.807** | **0.331** | **0.810** | **0.332** | **0.911** | **0.165** | **0.850** | **0.038** |

✗ indicates a module removed, and ✓ indicates a module added.

**Parameter Sensitivity** We study the parameter sensitivity of STAR, including the number of selected vectors in *Memory Router* ($K$), the total number of memory vectors ($N$), the rank of the adapter ($r$), and the hidden dimension size of the adapter ($d$). To enhance the flexibility of STAR, we search for the value of $d_{in}/r$ directly, rather than searching for $r$ itself. Figure 3a shows the impact of $K$. In most cases, a smaller $K$ is insufficient for effective learning. We typically select 7 as the preferred option. Besides, Figure 3b illustrates the impact of $N$. Common choices for this value are 15 or 25. This approach is more lightweight compared to defining a learnable vector for each state of each state variable (Shan et al., 2016). As shown in Figure 3c, by reducing the $d_{in}$ to $d_{in}/2$

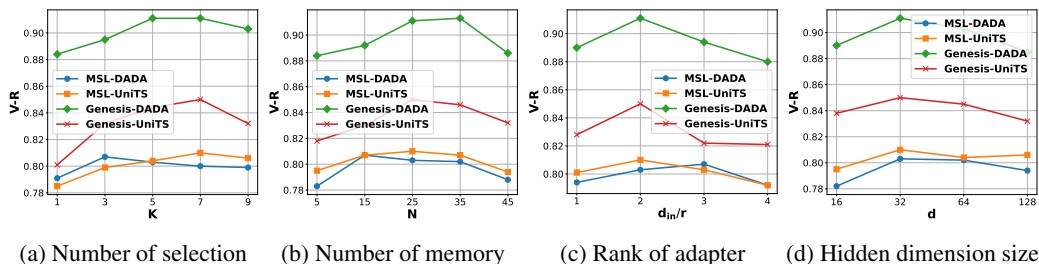

(a) Number of selection    (b) Number of memory    (c) Rank of adapter    (d) Hidden dimension size

Figure 3: Parameter sensitivity studies of main hyper-parameters in STAR.

or $d_{in}/3$, we can achieve a balance between efficiency and effectiveness. Figure 3d indicates that a large hidden dimension is not required, thus avoiding substantial additional computational costs.

**Efficiency Analysis** To further demonstrate that STAR introduces only a marginal performance overhead, we summarize the number of Parameters (M), Training time per epoch (s), and Inference time per sample (ms) for models using DADA and moment as backbones on the SMAP and SWAT datasets in the table below. Note that rows labeled '*' indicate that all parameters are fine-tuned, whereas the remaining configurations involve fine-tuning only the adapter parameters.

Table 4: Efficiency analysis.

| Dataset | SMAP | | | SWaT | | |
|---------|------|---|---|------|---|---|
| Metric | Parameters | Training time | Inference time | Parameters | Training time | Inference time |
| DADA* | 1.84 | 18.51 | 9.26 | 1.84 | 91.90 | 19.63 |
| +LoRA | 1.91 | 8.61 | 10.01 | 1.95 | 71.23 | 19.94 |
| +STAR | 2.00 | 10.44 | 11.64 | 2.11 | 78.92 | 20.11 |
| Moment* | 35.34 | 40.66 | 8.06 | 35.34 | 227.43 | 35.53 |
| +LoRA | 35.53 | 19.72 | 11.07 | 35.57 | 169.41 | 37.67 |
| +STAR | 35.92 | 22.66 | 11.60 | 36.01 | 198.87 | 40.91 |

**More Analysis** To demonstrate the applicability of STAR, we also evaluated its performance on datasets containing solely numerical variables (B.2). We further visualize the *Identity-guided State Encoder* to validate its effectiveness (B.3). Moreover, we present a case study to demonstrate the importance of state variables for anomaly detection (B.4).

## 5 CONCLUSION

In this study, we propose a novel STate-aware AdapteR (STAR) for TSFM in MTSAD. STAR is a plug-and-play module designed to address the failure of TSFMs to handle state variables during the fine-tuning stage. We propose the Identity-guided State Encoder and the Conditional Bottleneck Adapter to capture the complex semantics of states and the conditional influence of state variables, respectively. Our implementation is available at https://anonymous.4open.science/r/STAR-4540.

ETHICS STATEMENT

Our work exclusively uses publicly available benchmark datasets that contain no personally identifiable information. The proposed adapter for Time Series Foundation Models in Multivariate Time Series Anomaly Detection is designed for beneficial applications in system reliability and safety monitoring. No human subjects were involved in this research.

REPRODUCIBILITY STATEMENT

The performance of STAR and the datasets used in our work are real, and all experimental results can be reproduced. We have released our model code in an anonymous repository: https://anonymous.4open.science/r/STAR-4540. Once the paper is accepted, we will release the scripts for all settings.

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

# A EXPERIMENTAL DETAILS

## A.1 DATASETS

To evaluate our method, we selected five classic real-world multivariate anomaly detection datasets: 1) **MSL** (Hundman et al., 2018): Spacecraft incident and anomaly data from the MSL Curiosity rover. 2) **SMAP** (Hundman et al., 2018): Presents the soil samples and telemetry information used by the Mars rover. 3) **SWAT** (Mathur & Tippenhauer, 2016): Obtained from 51 sensors of the critical infrastructure system under continuous operations. 4) **Genesis** (von Birgelen & Niggemann, 2018): A sensor and control signals dataset collected from cyber-physical production systems. 5) **NYC** (Cui et al., 2016): The Transportation dataset provides information on taxi and ride-hailing trips in New York. Each of these datasets contains a significant number of *state variables*, with detailed statistics presented in Table 5.

Table 5: Statistics of multivariate datasets (AR: anomaly ratio).

| Dataset | Domain | Numeral Numbers | State Numbers | AR (%) | Avg Total Length | Avg Test Length |
|---------|--------|-----------------|---------------|--------|------------------|-----------------|
| MSL | Spacecraft | 1 | 54 | 5.88 | 132,046 | 73,729 |
| SMAP | Spacecraft | 1 | 24 | 9.72 | 562,800 | 427,617 |
| SWAT | Water treatment | 28 | 23 | 5.78 | 944,919 | 449,919 |
| Genesis | Machinery | 5 | 13 | 0.31 | 16,220 | 12,616 |
| NYC | Transport | 1 | 2 | 0.57 | 17,520 | 4,416 |

## A.2 DATA ANALYSIS

In the vast majority of cases, explicit input type information is unnecessary for the data processing. A significant difference between numerical variables and state variables in real-world scenarios lies in the number of their possible values. State variables are typically constrained to a limited number of outcomes, usually fewer than fifty. Conversely, numerical variables can assume a vast range of values, often numbering in the hundreds or thousands. This clear divergence in the size of their value space is sufficient to distinguish one type from the other.

The Table 6 presents a statistical summary of the value set cardinality for both numerical variables and state variables.

Table 6: Data Analysis.

| Dataset | Numerical Variables | | | | State Variables | | | |
|---------|---------------------|---------|--------|---------|-----------------|---------|--------|---------|
| Metric | Maximum | Minimum | Median | Average | Maximum | Minimum | Median | Average |
| MSL | 35103 | 35103 | 35103 | 35103.0 | 2 | 2 | 2.0 | 2.0 |
| SMAP | 75332 | 75332 | 75332 | 75332.0 | 2 | 2 | 2 | 2.0 |
| Genesis | 4074 | 512 | 2208 | 2250.8 | 21 | 2 | 2 | 3.3 |
| NYC | 11790 | 11790 | 11790 | 11790.0 | 48 | 2 | 7 | 19.0 |
| SWaT | 14626 | 116 | 2144 | 4002.6 | 37 | 2 | 2.0 | 3.9 |

## A.3 TIME SERIES FOUNDATION MODELS FOR ANOMALY DETECTION

In the realm of time series analysis, numerous models have surfaced in recent years. We choose models which designed for multi-task or specialised for anomaly detection, including task-specific model DADA (Shentu et al., 2024) and task-general models: UniTS (Gao et al., 2024), Moment (Goswami et al., 2024), Timer (Liu et al., 2024b). The specific descriptions for each of these models are listed in Table 7.

## A.4 METRICS FOR TIME SERIES ANOMALY DETECTION

The metrics used for evaluation can be divided into two categories: Threshold-relevant and Threshold-irrelevant. Threshold-relevant metrics depend on threshold parameters to convert

Table 7: Descriptions of Time Series Foundation Models for Anomaly Detection in experiments.

| Models | Descriptions |
|--------|--------------|
| DADA | DADA is a general time series anomaly detector pre-trained on multi-domain data, utilizing adaptive bottlenecks and dual adversarial decoders. It is designed to Detected by learning to differentiate between normal and abnormal patterns. |
| UniTS | UNITS is a unified, multi-task transformer model that uses task tokenization to handle a variety of time series tasks like forecasting, classification, imputation, and anomaly detection within a single framework. |
| Moment | Moment is a transformer system pre-trained on a masked time series task. It reconstructs masked portions of time series for tasks like forecasting, classification, anomaly detection, and imputation. |
| Timer | Timer is a GPT-style autoregressive model for time series analysis, predicting the next token in single-series sequences. It supports tasks like forecasting, imputation, and anomaly detection across different time series. |

anomaly scores into anomaly labels. Among them, Threshold-irrelevant metrics, on the other hand, evaluate the performance of the model based on the raw anomaly scores it generates. Please refer to Table 8 for an overview of these metrics.

Table 8: Overview of evaluation metrics.

| Category | Metric | Abbreviation | Short summary |
|----------|--------|--------------|---------------|
| **Threshold-relevant** | Accuracy | Acc | Measures the proportion of correct predictions among the total number of predictions. |
| | Range-Precision
Range-Recall
Range-F1-score | R-P Tatbul et al. (2018)
R-R Tatbul et al. (2018)
R-F1 Tatbul et al. (2018) | These metrics' definitions consider several factors: the ratio of detected anomaly subsequences to the total number of anomalies, the ratio of detected point outliers to total point outliers, the relative position of true positives within each anomaly subsequence, and the number of fragmented prediction regions corresponding to one real anomaly subsequence. |
| | Affiliated-Precision
Affiliated-Recall
Affiliated-F1-score | Aff-P Huet et al. (2022)
Aff-R Huet et al. (2022)
Aff-F1 Huet et al. (2022) | These metrics' definitions are the extension of the classical precision/recall/f1-score for time series anomaly detection that is local (each ground truth event is considered separately), parameter-free, and applicable generically on both point and subsequence anomalies. Besides the construction of these metrics makes them both theoretically principled and practically useful. |
| **Threshold-irrelevant** | AUC-PR | A-P Davis & Goadrich (2006) | Measures the area under the curve corresponding to Recall on the x-axis and Precision on the y-axis at various threshold settings. |
| | AUC-ROC
R-AUC-PR
R-AUC-ROC | A-R Fawcett (2006)
R-A-P Paparrizos et al. (2022)
R-A-R Paparrizos et al. (2022) | Measures the area under the curve corresponding to FPR on the x-axis and TPR on the y-axis at various threshold settings. They mitigate the issue that AUC-PR and AUC-ROC are designed for point-based anomaly detection, where each point is assigned equal weight in calculating the overall AUC. This makes them unsuitable for evaluating subsequence anomalies. |
| | VUS-PR
VUS-ROC | V-PR Paparrizos et al. (2022)
V-ROC Paparrizos et al. (2022) | VUS is an extension of the ROC and PR curves. It introduces a buffer region at the outliers' boundaries, thereby accommodating the false tolerance of labeling in the ground truth and assigning higher anomaly scores near the outlier boundaries. |

# B  MORE ANALYSIS ON STAR

## B.1  MORE HYPERPARAMETER ANALYSIS

To enhance the reproducibility and interpretability of the results, we conducted sensitivity analyses of $\lambda_1$ and $\lambda_2$. The VUS-ROC results for models using UniTS and DADA as backbones on the MSL and Genesis datasets are summarized in the Table 9.

As the results indicate, the model's performance is robust within a reasonable range for these hyperparameters. Optimal performance is consistently achieved when $\lambda_1$ is around $[5e-2, 5e-1]$ and $\lambda_2$ is around $[5e-3, 5e-2]$, which are the values we used for our main experiments.

Table 9: More hyperparameter analysis.

| Parameter | | MSL | | Genesis | |
|---|---|---|---|---|---|
| $\lambda_1$ | $\lambda_2$ | UniTS | DADA | UniTS | DADA |
| 1e-2 | 1e-3 | 0.801 | 0.790 | 0.844 | 0.905 |
| 5e-2 | 5e-3 | 0.808 | 0.799 | 0.847 | **0.911** |
| 1e-1 | 1e-2 | 0.807 | **0.803** | **0.850** | 0.908 |
| 5e-1 | 5e-2 | **0.810** | 0.796 | 0.845 | 0.907 |
| 1 | 1e-1 | 0.808 | 0.793 | 0.841 | 0.899 |
| 5 | 5e-1 | 0.802 | 0.791 | 0.840 | 0.897 |

## B.2  APPLICABILITY FOR DATASET

While most industrial scenarios feature a variety of *state variables*, there are also special or constrained settings where such variables are unavailable. To further extend the applicability of STAR, we explored the approach of proactively constructing pseudo-covariates to serve as *state variables* for processing by the STAR module. Specifically, numerical variables often inherently carry state information. For instance, temperature indicates whether the system is in a

Table 10: Evaluate STAR on datasets containing solely numerical variables.

| Dataset | | PSM | | | PUMP | | | ECG | |
|---|---|---|---|---|---|---|---|---|---|
| Metric | A-R | V-R | V-P | A-R | V-R | V-P | A-R | V-R | V-P |
| DADA | 0.606 | 0.581 | 0.403 | 0.775 | 0.778 | 0.226 | 0.839 | 0.807 | 0.508 |
| DADA* | 0.621 | 0.588 | 0.391 | 0.764 | 0.777 | 0.223 | 0.828 | 0.801 | 0.495 |
| +STAR* | **0.636** | **0.592** | **0.412** | **0.807** | **0.807** | **0.236** | **0.844** | **0.811** | **0.514** |
| UniTS | 0.582 | 0.573 | 0.377 | 0.415 | 0.573 | 0.211 | 0.872 | 0.840 | 0.522 |
| UniTS* | 0.548 | 0.544 | 0.354 | 0.384 | 0.536 | 0.195 | 0.863 | 0.835 | 0.516 |
| +STAR* | **0.607** | **0.602** | **0.401** | **0.436** | **0.598** | **0.217** | **0.884** | **0.848** | **0.537** |

high or low-temperature state, while the pH value reflects the precondition of an acidic or alkaline environment. We leverage this property by discretizing these numerical variables into discrete states to construct our pseudo-covariates. We conducted this experiment on the PSM Abdulaal et al. (2021) , PUMP Feng & Tian (2021), and ECG Yoon et al. (2020) datasets, and the results are presented in Table 10. Rows marked with an asterisk (*) denote the use of pseudo-covariates. The results indicate that in most cases, the incorporation of *state variables* still leads to a decrease in performance. Although the pseudo-covariates lack a well-defined physical meaning, STAR is nevertheless able to improve the performance of TSFMs to a certain degree.

## B.3  VISUALIZATION OF IDENTITY-GUIDED STATE ENCODER

To further validate the effectiveness of the *Identity-guided State Encoder*, we visualized its output embeddings as well as the selections made by the *Memory Router*. We selected the Genesis (von Birgelen & Niggemann, 2018) for our analysis, and a subset of its variables is described in Table B.3.

We use $Id_v$ to denote the variable identity (index of *state variable*), and $Id_s$ to denote the state identity (discrete value of *state variable*). Figure 4a illustrates the similarity between embeddings of different variables, where a distinct clustering pattern is observed. This indicates that variables with similar semantic information exhibit higher similarity. Since our router is conditioned on both variable identity and state identity, embeddings for variables that share the same variable identity or state identity exhibit greater similarity. However, the overall influ-

Table 11: Evaluate STAR on datasets containing solely numerical variables.

| $Id_v$ | Group | Variable name | Descriptions |
|---|---|---|---|
| 1 | Silder State | Slider_OUT | The slider is on the outside. |
| 2 | | Slider_IN | The slider is on the inside. |
| 3 | Detection State | NonMetall | Non-metal detected. |
| 4 | | Metall | Metal detected. |
| 5 | Position State | Pos4reached | Arrived at Position 4. |
| 6 | | Pos3reached | Arrived at Position 3. |
| 7 | | Pos2reached | Arrived at Position 2. |
| 8 | | Pos1reached | Arrived at Position 1. |

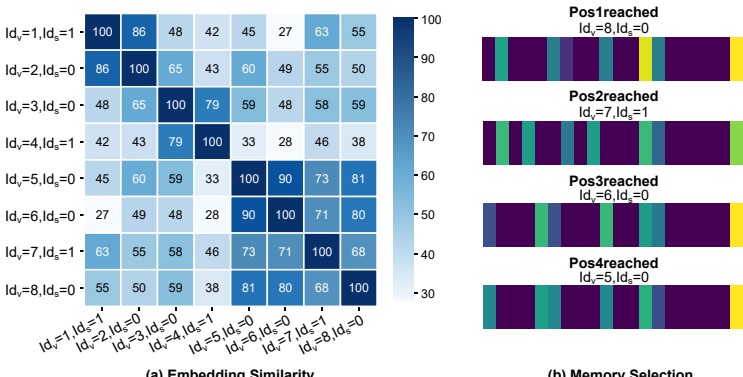

Figure 4: (a) shows the similarity between the embeddings output from *Identity-guided State Encoder*. (b) shows the weight of selection in *Memory Router*.

ence of variable identity is more pronounced than that of state identity. These observations are further corroborated by Figure 4b.

## B.4 CASE STUDY

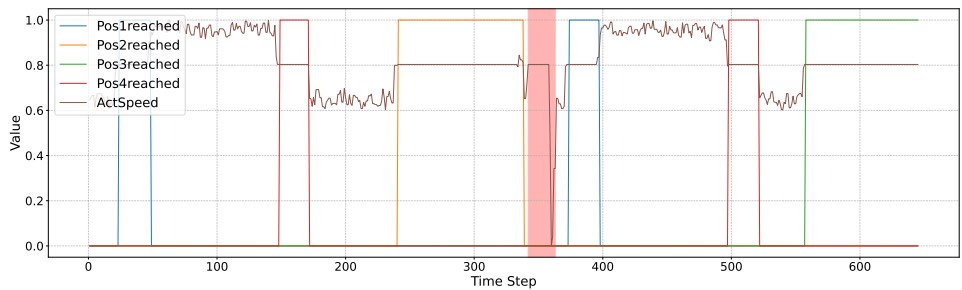

Figure 5: Case series in Genesis.

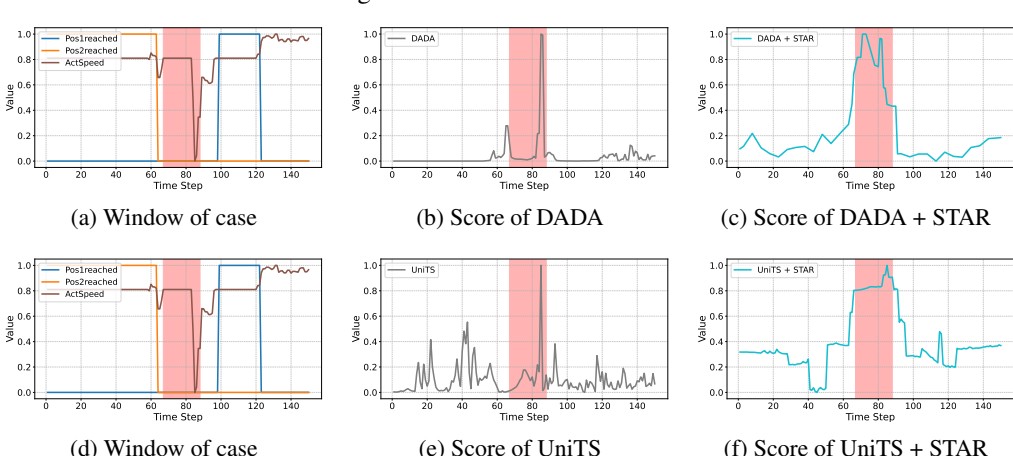

(a) Window of case      (b) Score of DADA      (c) Score of DADA + STAR

(d) Window of case      (e) Score of UniTS      (f) Score of UniTS + STAR

Figure 6: Anomaly score of backbones and STAR in case from Genesis.

To further analyze how STAR works, we selected and visualized cases from real-world datasets.

(1) **Genesis** Figure 5 illustrates that the position state variables (Pos1reached, Pos2reached, Pos3reached, and Pos4reached) have a condition-based inference on the actual motor speed (ActSpeed). Under normal circumstances, the actual motor speed exhibits a stable pattern when the system is in a state associated with reaching a specific position (PosXreached = 1). However, when the system is not reached at any specific position, the motor speed displays an unstable pattern (PosXreached = 0).

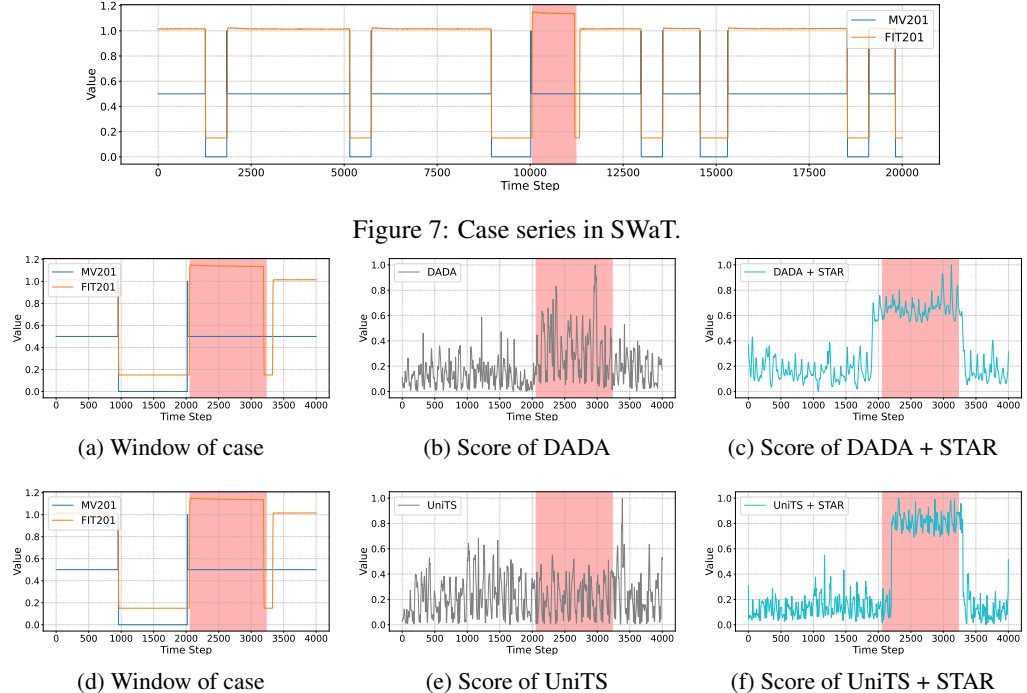

Figure 7: Case series in SWaT.

(a) Window of case

(b) Score of DADA

(c) Score of DADA + STAR

(d) Window of case

(e) Score of UniTS

(f) Score of UniTS + STAR

Figure 8: Anomaly score of backbones and STAR in case from SWaT.

The anomaly presented in the case is that the actual motor speed exhibits a stable pattern even when the system is in a state of not having reached any specific position. This constitutes a state-value mismatch anomaly. As shown in Figure 6, although TSFMs are highly sensitive to the abrupt change at the final timestamps of the anomaly, they fail to detect the preceding portion. With the integration of STAR, the model gains heightened sensitivity to the state-value mismatch anomaly and successfully captures this anomaly case.

(2) **SWaT** As shown in Figure 5, the state variable of motorized valve switch (MV201) also has a condition-based inference on the water flow (FIT201). Under normal conditions, the behavior is as follows: when the valve is open (MV201=0.5), the water flow remains stable at a high level; when the valve is closed (MV201=0), the flow stabilizes at a low level; and during a transitional state (MV201=1), the flow rapidly increases or decreases.

The anomaly presented in the case is twofold. On the one hand, the water level is excessively high. On the other hand, the flow begins to decrease rapidly even though the valve is not in a transitional state (MV201=1), remaining fully open (MV201=0.5). Although DADA successfully identified the excessively high water level, it failed to detect the anomaly of the rapid decrease. This is likely because the variable itself frequently exhibits patterns of rapid decline over short periods. STAR successfully captures this anomaly by considering the degree of matching between the state and the numerical value.

## B.5 VISUALISATION OF CASE IN SMAP

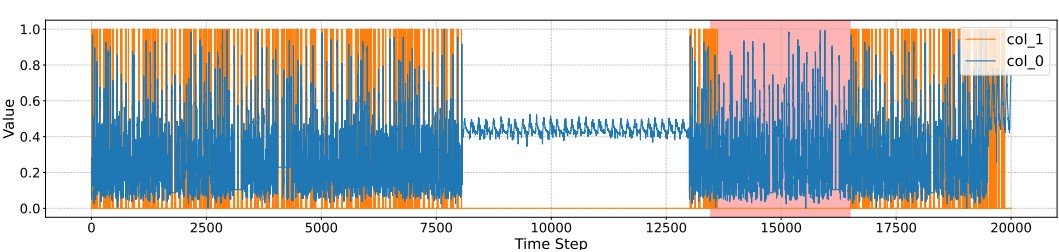

Figure 9: Case series in SMAP.

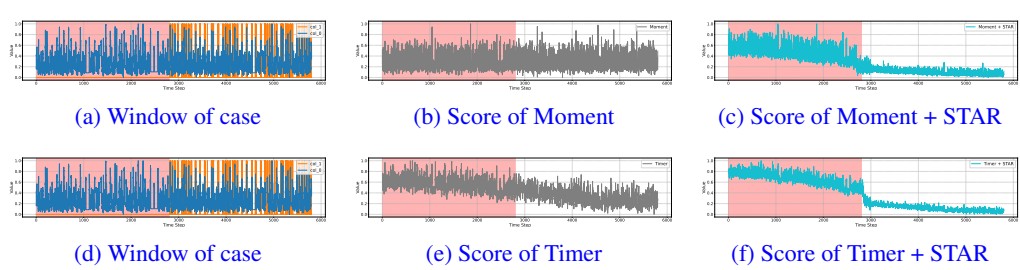

| (a) Window of case | (b) Score of Moment | (c) Score of Moment + STAR |
| (d) Window of case | (e) Score of Timer | (f) Score of Timer + STAR |

Figure 10: Anomaly score of backbones and STAR in case from SMAP.

Figure 9 illustrates the context of an anomalous case, highlighting a clear conditional relationship between the state and numerical variables under normal conditions. Specifically, when the state variable (col_1) fluctuates, the numerical variable (col_0) exhibits high-amplitude oscillations; conversely, when the state variable is stable, the numerical variable maintains a relatively stable pattern.

As is evident in the Figure 10, without STAR, both the Moment and Timer models exhibit poor discrimination for this anomalous case, suffering from a degree of false alarms (false positives) and missed detections (false negatives). In contrast, with the integration of STAR, the anomaly scores provide a much clearer distinction, successfully reducing missed detections in the left region and mitigating false alarms in the right.

## B.6 VISUALISATION OF BOTTLENECK ACROSS DIFFERENT STATE

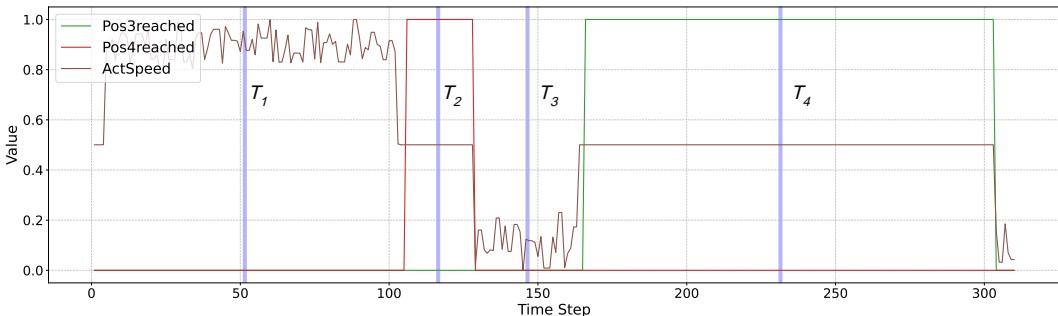

Figure 11: Case series in Genesis.

To analyze the bottleneck activations under different states, we visualize both the raw time series data and the corresponding mask vector. This mask vector is conditioned on state variables and applied to matrix $R$, which is inserted between matrices $A$ and $B$ in the Conditional Bottleneck Adapter to adaptively modulate the bottleneck, as defined in Equation 9.

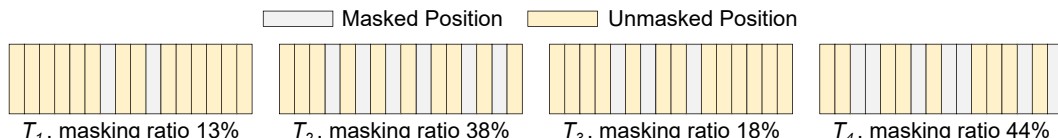

Figure 12: Visualization of the mask vector of $\boldsymbol{R}$.

Figure 11 illustrates the raw time series, highlighting the relationship between numerical and state variables in the Genesis dataset. When all state variables (Pos3reached and Pos4reached) are zero, the numerical variable (ActSpeed) exhibits complex and rapid fluctuations; otherwise, it displays a stable pattern. We specifically selected representative timestamps $T_1$ and $T_3$ to exemplify the rapid fluctuation pattern, and $T_2$ and $T_4$ for the relatively stable pattern, as indicated by the purple markers in the figure.

Figure 12illustrates the mask vector of matrix $\boldsymbol{R}$ corresponding to timestamps $T_1$, $T_2$, $T_3$, and $T_4$, along with its masking ratio. This masking ratio reflects the bottleneck size of the weights in the Conditional Bottleneck Adapter.

Timestamps $T_1$ and $T_3$ exemplify the rapid fluctuation pattern. In these instances, the numerical variable is relatively complex and requires more temporal information for reconstruction. Consequently, the masking ratio of the corresponding matrix $\boldsymbol{R}$ is lower, indicating a larger bottleneck size. In contrast, $T_2$ and $T_4$ represent the stable pattern, where the numerical variable is relatively simple and requires less temporal information. This results in a higher masking ratio for $\boldsymbol{R}$, signifying a smaller bottleneck size.

This phenomenon justifies our strategy of dynamically adjusting the bottleneck size according to the system state.

## C  THE USE OF LARGE LANGUAGE MODELS

The use of open-source Large Language Models (LLMs) in this work was strictly limited to assisting with the translation of certain terms and polishing a small portion of the text. LLMs did not contribute to the conceptual aspects of the research, including information retrieval, knowledge discovery, or the ideation process.

