# OpenReview forum: "STAR: Boosting Time Series Foundation Models for Anomaly Detection  Through State-aware Adapter"
_ICLR.cc/2026/Conference — Submitted to ICLR 2026_

### Official Review · Reviewer_cWBj · 2025-10-23

**Soundness:** 3
**Presentation:** 3
**Contribution:** 3
**Rating:** 6
**Confidence:** 5

**Summary:**

This paper proposes a novel approach called STAR (STate-aware AdapteR) to enhance Time Series Foundation Models (TSFMs) for Multivariate Time Series Anomaly Detection, particularly in real-world industrial scenarios where time series consist of both numerical variables and discrete state variables that describe system statuses. Specifically, STAR introduces three core components: first, the Identity-guided State Encoder, which captures the complex semantics of state variables through a learnable state memory; second, the Conditional Bottleneck Adapter, which dynamically generates low-rank adaptation parameters based on the current state, allowing for flexible incorporation of state information into the TSFM; and third, the Numeral-State Matching module, which uses contrastive learning to improve anomaly detection for both numerical and state variables. Experimental results on several real-world datasets demonstrate that STAR significantly improves the performance of existing TSFMs in anomaly detection.

**Strengths:**

S1. The paper addresses a significant gap in existing Time Series Foundation Models (TSFMs) by recognizing the importance of state variables, which are often overlooked or treated uniformly with numerical variables. By introducing STAR (STate-aware AdapteR), the authors propose a novel method that more effectively handles categorical state variables and their conditional influence on numerical variables, improving anomaly detection in complex real-world scenarios.

S2. STAR is designed as a plug-and-play module that can be seamlessly integrated into existing TSFMs during the fine-tuning stage. This makes it a practical solution for a wide range of applications without the need for completely reworking existing models.

S3. The paper demonstrates the effectiveness of STAR through extensive experiments on multiple real-world datasets with state variables. The results consistently show that STAR improves the performance of TSFMs, particularly in complex scenarios where state variables are essential.

**Weaknesses:**

W1. While the paper includes an ablation study to evaluate the performance of different modules in STAR, it lacks an experiment that isolates the Identity-guided State Encoder (ID-SE) and replaces it with a simpler model like an MLP. Analyzing this would help to better understand the specific contribution of the ID-SE and whether its complexity is justified or if a simpler approach could yield comparable results, which could provide deeper insights into the trade-offs in model design.

W2. The paper does not mention some contrastive-based methods such as  TFMAE[1] and CTAD[2].

W3. There is a minor typographical error in line 270, where "R, B" is used, but it should be "R, D." Such typographical mistakes can cause confusion, especially in technical papers. Ensuring the accuracy of notation and consistently following the same terminology would enhance the paper’s clarity and professionalism.

[1] Temporal-Frequency Masked Autoencoders for Time Series Anomaly Detection

[2] Contrastive Time-Series Anomaly Detection

**Questions:**

See weaknesses.

---

> ### Author Response · Authors · 2025-11-21
>
> Dear Reviewer cWBj, thank you for providing your detailed and constructive feedback.
>
> ---
> ### Response to W1:
>
> We sincerely thank you for this insightful suggestion. To further validate the effectiveness of the ID-SE, we conduct the suggested ablation study. We replaced the ID-SE module with a standard MLP-based encoder, which serves as a simpler baseline for encoding state variables. The VUS-ROC results for this comparison are summarized in the table below.
>
> |Dataset||||MSL||Genesis||
> |:-:|:-:|:-:|:-:|:-|:-:|:-:|:-:|
> |**ID-SE**|**MLP**|**CB Adapter**|**N-S Matching**|**DADA**|**UniTS**|**DADA**|**UniTS**|
> |✔️|❌|✔️|❌|**0.803**|**0.807**|**0.872**|**0.819**|
> |❌|✔️|✔️|❌|0.751|0.799|0.849|0.793|
> |✔️|❌|❌|✔️|**0.760**|**0.805**|**0.854**|**0.805**|
> |❌|❌|❌|✔️|0.742|0.796|0.844|0.794|
> |✔️|❌|✔️|✔️|**0.807**|**0.810**|**0.911**|**0.850**|
> |❌|✔️|✔️|✔️|0.763|0.803|0.866|0.824|
>
>
> ---
> ### Response to W2:
> We sincerely thank you for pointing out these relevant and important works.
> While TFMAE and CTAD also leverage contrastive learning, they create contrastive pairs by augmenting the numerical signal itself (e.g., TFMAE through time-frequency transformations, and CTAD via noise injection) to learn a robust representation of normal patterns. In contrast, our method formulates the contrast based on the intrinsic relationship between two different data modalities present in the data: the numerical variables and the state variables. Our goal is to explicitly learn the degree of correspondence between these two types of variables.
> We also include relevant descriptions in the manuscript (Lines 340-342).
>
> ---
> ### Response to W3:
> We thank you for your careful attention to detail and for identifying this typo. We correct '$\boldsymbol{R}, \boldsymbol{B}$' to '$\boldsymbol{R}, \boldsymbol{D}$' in the manuscript and performed a full review of the paper to check for notational consistency throughout.
>
> ---
>
> **Thanks for your sincere suggestions again! If you have any additional question, we can make further discussion!** 😊

---

> > ### Comment · Reviewer_cWBj · 2025-11-26
> >
> > Thank you for the clarifications and the additional evidence provided during the rebuttal. The new analysis and results effectively addressed my earlier concerns and substantiated the performance gains attributable to the proposed method. I will raise my score accordingly.

---

> > > ### Author Response · Authors · 2025-11-26
> > >
> > > Dear Reviewer cWBj:
> > >
> > > Thank you so much for your encouraging feedback and for your support toward the acceptance of our paper. We sincerely appreciate your time and constructive comments throughout the review process.
> > >
> > > Best regards,
> > >
> > > Authors

---

### Official Review · Reviewer_taXc · 2025-10-24

**Soundness:** 2
**Presentation:** 2
**Contribution:** 4
**Rating:** 6
**Confidence:** 4

**Summary:**

This paper addresses the difficulty of the TSFM model in effectively handling discrete state values during anomaly detection. It proposes an efficient method for representing discrete state–type numerical features and using them as conditions in a conditional information bottleneck module to fine-tune the TSFM model.

**Strengths:**

1. This paper addresses an important practical problem encountered in time-series anomaly detection models — how to effectively handle discrete state variables.

2. Overall, the paper is logically coherent, clearly presented, and well-structured.

3. Experimental results demonstrate that the proposed method effectively improves the accuracy of anomaly detection in the TSFM model.

4. The design that employs an MoE-like structure for state representation, along with the integration of LoRA and the conditional information bottleneck within the conditional network, exhibits a certain degree of novelty.

**Weaknesses:**

1.The description of matrix operations such as aggregating, averaging, and variance computation along a specific dimension is unclear. Readers must infer which dimension the operation applies to based on the data shape, which makes the paper somewhat difficult to follow.

2.When applying the conditional information bottleneck, the loss function does not include any mutual information optimization term related to this network. This raises concerns about whether the method can reliably ensure that the conditional information bottleneck compresses irrelevant information while preserving relevant information.

3.A key but insufficiently discussed concept in the paper is the distinction between variable identities and state identities. The authors present this distinction as one of the core innovations and contributions, yet they do not clearly define what state identities and variable identities mean. Since these are not standard terms in the field, they require detailed clarification.

**Questions:**

1. Could you please discuss why the conditional information bottleneck can effectively preserve the relevant information while compress irrelevant ones without the mutual information loss term?

2. Could you please discuss what the variable identities and state identities mean?

---

> ### Author Response · Authors · 2025-11-21
>
> Dear Reviewer taXc, thank you for providing your detailed and constructive feedback.
>
> ---
> ### Response to W1:
>
> We thank you for this helpful comment. We have addressed the ambiguity regarding matrix operation dimensions in the revised manuscript.
>
> Specifically, we clarify the descriptions and notation for Equations (4), (5), (7), and (16) to make the operational dimensions explicit. We hope these revisions resolve your concern and improve the paper's overall clarity.
>
> ---
>
> ### Response to W2&Q1:
> We sincerely thank you for this deep and insightful question.
>
> The issue regarding the 'conditional information bottleneck' may be due to our imprecise use of terminology, which caused you confusion.
>
> To clarify, the 'bottleneck' we referred to denotes the bottleneck size of the weights used for the reconstruction task, rather than the Information Bottleneck in the standard sense.
>
> To avoid this misunderstanding, we have revised the manuscript and replaced the term 'information bottleneck' with 'bottleneck size of the weights' in the paper (Lines 287,304). In our method, this bottleneck regulates the degree of feature compression for the reconstruction task and can be directly supervised by the reconstruction loss.
>
> We also provide relevant visualizations in Appendix B.6 (Lines 1062-1104), to further demonstrate its effectiveness. We hope that this clarification, along with the supplementary materials, will address your concerns.
>
> ---
>
> ### Response to W3&Q2:
> We sincerely thank you for this critical feedback.
>
> In our original manuscript, we did provide a brief definition for these terms in lines 194-196, which states: "... variable identities (**indices of state variables**) and the state identities (**discrete values of state variables**) ..."
>
> To clarify this point further, we have renamed these terms (**variable identity** ➜ **state identity**, **state identity** ➜ **state value**) to avoid ambiguity and provide a more detailed explanation of these terms below:
>
> The Identity-guided State Encoder processes each state variable to capture its information from both the **state identity** (i.e., index of the state variable) and the **state value** (i.e., discrete value of the state variable) at each time point.
>
> Specifically, the **state identity** serves as the unique, static identifier for a state variable within the multivariate time series, corresponding directly to its channel index. For example, a variable located at index $i$ has a variable identity of $i$.
>
> In contrast, the **state value** is the discrete value of a state variable at a specific time point. For instance, if a state variable has a value of $j$ at a given time step, its state value is $j$. Note that these values are mapped to non-negative integers (e.g., \{0, 1, 2, ...\}) during data preprocessing.
>
>
> We have included this explanation in the revised manuscript (Lines 198-204).
>
> **Thanks for your sincere suggestions again! If you have any additional questions, we can have further discussions!** 😊

---

> > ### Comment · Reviewer_taXc · 2025-11-26
> >
> > Thank you for your response. The authors have dealed with my concerns.

---

> > > ### Author Response · Authors · 2025-11-26
> > >
> > > Dear Reviewer taXc:
> > >
> > > We are pleased to learn that we have successfully addressed your concerns. We kindly ask if you could consider raising the score, as this would give us a better chance of presenting our work at the conference. We would be deeply grateful. Thank you again for taking the time to review and comment on our paper.
> > >
> > > Best regards,
> > >
> > > Authors

---

### Official Review · Reviewer_3Yzz · 2025-10-28

**Soundness:** 2
**Presentation:** 3
**Contribution:** 2
**Rating:** 4
**Confidence:** 3

**Summary:**

The paper proposes STAR (STate-aware AdapteR), a plug-and-play module to enhance TSFMs for multivariate time series anomaly detection involving both numerical and discrete state variables. STAR includes (1) an identity-guided state encoder for categorical semantics, (2) a conditional bottleneck adapter for state-conditioned adaptation, and (3) a numeral–state matching module for state-related anomaly detection. Experiments on real-world datasets show that STAR improves TSFM performance.

**Strengths:**

The paper presents a clear and original perspective by emphasizing the overlooked distinction between numerical and discrete state variables in multivariate time series anomaly detection. This problem formulation is both valid and practically significant, as many real-world industrial systems contain mixed-variable types. The proposed STAR module is a well-motivated and technically sound solution that effectively integrates state-aware adaptation into existing TSFMs. The experimental evaluation is comprehensive and convincing, covering multiple real-world datasets and various TSFM backbones, consistently demonstrating performance improvements.

**Weaknesses:**

1. The model complexity deserves further discussion. Compared with lightweight adaptation methods such as LoRA, the proposed STAR introduces additional components (state encoder, conditional adapter, and matching module), which may increase the computational cost and implementation complexity.
2. The issue of handling numerical versus discrete state variables, while valid, might represent a relatively minor challenge within TSFM-based anomaly detection. Theoretically, a sufficiently generalizable TSFM could already accommodate such heterogeneity without explicit modeling. From the experimental results, the performance gains are not always substantial, particularly for combinations such as Moment + SMAP and Timer + SMAP. Therefore, whether such a design trade-off is justified in real-world deployment scenarios warrants further consideration.
3. The experimental implementation details are somewhat incomplete. For instance, in line 329, the settings and balancing strategies for the hyperparameters λ₁ and λ₂ are not clearly explained. A more detailed discussion would enhance the reproducibility and interpretability of the results.

**Questions:**

Does the proposed method require explicit input type information, i.e., predefined knowledge of which channels are numerical variables and which are discrete state variables? If so, how sensitive is the performance to potential misclassification or ambiguity of variable types?

---

> ### Author Response · Authors · 2025-11-21
>
> Dear Reviewer 3Yzz, thank you for providing your detailed and constructive feedback.
>
> ---
> ### Response to W1:
> Our method only introduces an additional computational cost that is linear with respect to the input sequence length $T$ and the number of state variables $C_s$. We list all the theoretical complexities of STAR and backbones in the following table:
>
> |Method|STAR|DADA|UniTS|Moment|Timer|
> |:-:|:-:|:-:|:-:|:-:|:-:|
> |**Complexity**|$O(T·C_s·d)$|$O(C·T^2·d)$|$O(C·T^2·d + T·C^2·d)$|$O(C·T^2·d)$|$O(C·T^2·d)$|
>
> Where $C$ is the number of variables input to the backbone and $d$ is the size of the hidden dimension.
>
> To further demonstrate that STAR introduces only a marginal performance overhead, we summarize the number of parameters (M), training time per epoch (s), and inference time per sample (ms) for models using DADA and Moment as backbones on the SMAP and SWAT datasets in the table below. Note that rows labeled '(Full fine-tuning)' indicate that all parameters are fine-tuned, whereas the remaining configurations involve fine-tuning only the adapter parameters.
>
> |Dataset|SMAP|||SWaT|||
> |:-|:-|:-|:-|:-|:-|:-|
> |**Metric**|**Parameters**|**Training time**|**Inference time**|**Parameters**|**Training time**|**Inference time**|
> |**DADA** (Full fine-tuning)|1.84|18.51|9.26|1.84|91.90|19.63|
> |$~+$ **LoRA** (Fine-tuning)|1.91|8.61|10.01|1.95|71.23|19.94|
> |$~+$ **STAR** (Fine-tuning)|2.24|10.44|11.64|2.11|78.92|20.11|
> |**Moment** (Full fine-tuning)|35.34|40.66|8.06|35.34|227.43|35.53|
> |$~+$ **LoRA** (Fine-tuning)|35.53|19.72|11.07|35.57|169.41|37.67|
> |$~+$ **STAR** (Fine-tuning)|36.08|22.66|11.60|36.01|198.87|40.91|
>
> The results demonstrate that our method outperforms LoRA (Table 1, Lines 432–449) while maintaining comparable efficiency, achieving faster training speeds than full fine-tuning with only minimal inference overhead.
>
>
> We have also incorporated this efficiency analysis into our manuscript (Lines 500–514).
>
>
> ---
>
> ### Response to W2:
> We would like to clarify that the issue of handling discrete state variables is **not** a relatively minor challenge. In fact, the topic of our paper comes from real industrial applications, where significant amount of state variables exist, such as water treatment scenario. By defining the system's operational context, these variables directly govern the behavior of the numerical variables, including their amplitude and evolution over time.
>
> But state variables vary significantly across different datasets in terms of the total number of variables, the number of states for each variable, and their specific mechanisms of influence. This significant differences pose a substantial challenge for TSFMs to achieve robust generalization for these variables during pre-training. Consequently, while a sufficiently generalizable TSFM may be theoretically plausible, **existing TSFM methods demonstrably lack this capability in practice**, as illustrated in Figure 1c (Lines 54-64).
>
> This gap motivates us to design a solution at the fine-tuning stage, which enables better modeling of state variables for specific scenarios.
>
> Although the performance gains on certain metrics may be marginal in a few cases, our method's primary contribution is to **equip the backbone with a more comprehensive detection capability at a minimal performance cost**. Specifically, it enables the model to leverage system state information to **capture anomalies to which the backbone is inherently insensitive**, as well as to **reduce false alarms**, as shown in the newly added Appendix B.5 (Lines 1028-1059).
>
> In summary, the motivation for our model stems directly from tangible challenges in real-world industrial applications, where existing TSFMs fail to handle effectively. This ensures the practical relevance and value of our work in such scenarios.

---

> ### Author Response · Authors · 2025-11-21
>
> ### Response to W3:
> To enhance the reproducibility and interpretability of the results, we conducted sensitivity analyses of λ₁ and λ₂. The VUS-ROC results for models using UniTS and DADA as backbones on the MSL and Genesis datasets are summarized in the table below.
> |Hyperparameter||MSL||Genesis||
> |-:|:-:|:-|:-:|:-|:-:|
> |**λ₁**|**λ₂**|**UniTS**|**DADA**|**UniTS**|**DADA**|
> |1e-2|1e-3|0.801|0.790|0.844|0.905|
> |5e-2|5e-3|0.808|0.799|0.847|**0.911**|
> |1e-1|1e-2|0.807|**0.803**|**0.850**|0.908|
> |5e-1|5e-2|**0.810**|0.796|0.845|0.907|
> |1|1e-1|0.808|0.793|0.841|0.899|
> |5|5e-1|0.802|0.791|0.840|0.897|
>
> As the results indicate, the model's performance is robust within a reasonable range for these hyperparameters. Optimal performance is consistently achieved when λ₁ is around [5e-2, 5e-1] and λ₂ is around [5e-3, 5e-2], which are the values we used for our main experiments.
>
> We have also incorporated this efficiency analysis into our manuscript (Lines 866-878).
>
> ---
>
> ### Response to Q1:
> In the vast majority of cases, explicit input type information is unnecessary for the data processing.
>
> A significant difference between numerical variables and state variables in real-world scenarios lies in the number of their possible values. State variables are typically constrained to a limited number of outcomes, usually fewer than fifty. Conversely, numerical variables can assume a vast range of values, often numbering in the hundreds or thousands. This clear divergence in the size of their value space is sufficient to distinguish one type from the other.
>
> The following table presents a statistical summary of the value set cardinality for both numerical variables and state variables.
>
> |Dataset|Numerical|variables|||State|variables|||
> |:-|:-:|:-:|:-:|:-|:-:|:-:|:-:|:-:|
> |**Metric**|**Maximum**|**Minimum**|**Median**|**Average**|**Maximum**|**Minimum**|**Median**|**Average**|
> |**MSL**|35103|35103|35103|35103.0 |2|2|2.0|2.0
> |**SMAP**|75332|75332|75332|75332.0 |2|2|2|2.0
> |**Genesis**|4074|512|2208|2250.8 |21|2|2|3.3
> |**NYC**|11790|11790|11790|11790.0 |48|2|7|19.0
> |**SWaT**|14626|116|2144|4002.6 |37|2|2.0|3.9
>
>
> We have also incorporated this analysis into our manuscript (Lines 779-797).
>
> ---
>
> **Thanks for your sincere suggestions again! If you have any additional questions, we can have further discussions!** 😊

---

### Official Review · Reviewer_oaB6 · 2025-11-01

**Soundness:** 2
**Presentation:** 2
**Contribution:** 2
**Rating:** 2
**Confidence:** 3

**Summary:**

The paper addresses a limitation of existing Time Series Foundation Models (TSFMs), which typically overlook the state variables associated with time series data. The authors propose integrating the learning of state variables into TSFMs and explore several architectural designs to optimise this integration.

**Strengths:**

•	The paper provides a clear rationale for why careful model design is necessary to effectively leverage state variables.

•	Experimental evidence supports the authors’ claim that naively unifying state-variable modelling within TSFMs does not improve performance.

•	The adapted STAR approach demonstrates performance gains, supporting the value of the proposed design.

**Weaknesses:**

•	The novelty of some technical components is limited. Several methods are described as being inspired by prior work (e.g., LoRA and Shentu et al., 2024), but the precise distinctions from these works are not clearly explained.

•	Certain technical details remain ambiguous. For instance, how do dynamic masks work in your model?

•	Some variables in the equations are undefined, making parts of the formulation difficult to follow.

•	The paper lacks runtime or complexity analysis. Since the CB Adapter generates a new matrix R and vector D for each patch, quantifying the associated computational overhead would make the efficiency claims more convincing.

**Questions:**

- In Eq. (1) (lines 213–215), not all notations are defined. For example, it is unclear whether i denotes the variable index or the time index (assumed to be the variable index).

- In Eq. (5) (lines 234–235), the second denominator avg(E_sel) may be a typo and should possibly read avg(E_imp) instead.

- How is the dynamic mask incorporated into the proposed method? The current description only provides formulas without an intuitive explanation. A short textual or diagrammatic illustration would help.

- Both Figure 2 (lines 162–182) and Section 3.2 (lines 265–289) show the CB Adapter applied to a generic weight matrix W_0, but it is unclear to which specific layers this adapter is attached. Please clarify where within the model architecture the CB Adapter is inserted.

- Appendix B.2 presents a t-SNE example. Including additional visualisations, for example, adapter activations or differences in anomaly scores across states, would provide deeper insight into how state information influences model behaviour.

---

> ### Author Response · Authors · 2025-11-21
>
> Dear Reviewer oaB6, thank you for providing your detailed and constructive feedback.
>
> ---
> ### Response to W1: Lack of comparison with previous works
>
> **Compared with LoRA:** While our method also employs a low-rank design, it fundamentally diverges from the standard LoRA framework.
>
> Standard LoRA consists of two static low-rank matrices ($\boldsymbol{A}$ and $\boldsymbol{B}$) trained from scratch. In contrast, the core novelty of our approach lies in generating dynamic parameters conditioned on state variables. Specifically, we derive matrices $\boldsymbol{A}$ and $\boldsymbol{B}$ via decomposition and keep them frozen, while the training process focuses exclusively on the dynamic generation of matrices $\boldsymbol{R}$ and $\boldsymbol{D}$ to modulate the overall weights. Our method achieves state-awareness and retains the same versatile applicability as standard LoRA.
>
> |Method|Low-rank|Dynamic parameters|State-awareness|Training method|
> |:-:|:-:|:-:|:-:|:-|
> |LoRA|✔️|❌|❌|$\boldsymbol{A}$ and $\boldsymbol{B}$ trained from scratch|
> |Ours|✔️|✔️|✔️|$\boldsymbol{A}$ and $\boldsymbol{B}$: frozen, $\boldsymbol{R}$ and $\boldsymbol{D}$: generated|
>
> **Compared with Shentu et al., 2024:** Although both methods deal with dynamic bottleneck sizes, the underlying mechanisms are fundamentally different.
>
> Their method employs a Mixture of Experts (MoE) architecture, utilizing a routing mechanism to dynamically select bottleneck sizes. In contrast, our approach leverages state variables to generate an adaptive mask vector, thereby dynamically controlling the bottleneck sizes via masking the $\boldsymbol{R}$ matrix.
>
> Their approach is primarily designed as a foundation model, where the bottleneck size selection approach is trained during the pre-training phase, without the necessity for reducing the parameter count or counting for efficiency. In contrast, our implementation is trained during the fine-tuning stage and tailored for lightweight adapters, thereby achieving higher parameter efficiency. Furthermore, while their mechanism mainly leverages temporal information, our method places greater emphasis on system state information encapsulated within state variables.
>
>
> |Method|Dynamic bottleneck|Adaptive method|Lightweight|State-awareness|
> |:-:|:-:|:-|:-:|:-:|
> |Shentu et al.|✔️|Mixture of Experts|❌|❌|
> |Ours|✔️|Dynamic Mask|✔️|✔️|
>
> ---
> ### Response to W2&Q3,4: Unclear description of technical details
> We sincerely thank you for your insightful feedback and for highlighting these ambiguities. We provide relevant clarifications as follows:
>
> **Use of Dynamic masks:** Thank you for your suggestion. We have added the following content in our manuscript (Lines 313-316) to improve the clarity of the manuscript:
>
> As shown in Figure 2b (Lines 162–182), we adaptively learn a state-aware dynamic mask vector based on current state variables via a soft-masking mechanism. We apply this mask to the matrix $\boldsymbol{R}$ and then insert this matrix between the low-rank matrices $\boldsymbol{A}$ and $\boldsymbol{B}$ to modulate the bottleneck size of the Conditional Bottleneck (CB) Adapter, as defined in Eq. 9.
>
> **Use of CB Adapter:**
> This adapter is designed to possess general-purpose capabilities. Theoretically, it can be integrated with the linear layers of any module. In practice, we apply it to the input layer, the output layer, and specific layers in the encoder or the decoder (e.g., the k, q, and v matrices of the attention layers). We have incorporated this clarification into the revised manuscript (Lines 291-293).
>
> ---
> ### Response to W3&Q1,2: Writing errors
> Thank you very much for your detailed review and valuable feedback. Your insightful comments are instrumental in improving the quality of our manuscript. We have revised the manuscript to incorporate these corrections:
>
> **For Eq. (1):** We clarify that $i$ denotes the variable index, as you correctly assumed. This clarification has been added immediately following Eq. (1) (Line 222).
>
> **For Eq. (5):** The term "avg($E_{sel}$)" in the denominator is indeed a typo and should be "avg($E_{imp}$)". We have corrected this in the revised version of Eq. (5) (Lines 243-248).
>
> We have carefully reviewed the manuscript and corrected this and other similar errors, such as another typo in Eq. (1). Thank you again for your sharp observations, which have helped us significantly improve the clarity of our paper.

---

> ### Author Response · Authors · 2025-11-21
>
> ### Response to W4: Runtime and complexity analysis
> We thank you for this valuable feedback.
> Our method only introduces an additional computational cost that is linear with respect to the input sequence length $T$ and the number of state variables $C_s$. We list all the theoretical complexities of STAR and backbones in the following table:
>
> |Method|STAR|DADA|UniTS|Moment|Timer|
> |:-:|:-:|:-:|:-:|:-:|:-:|
> |**Complexity**|$O(T·C_s·d)$|$O(C·T^2·d)$|$O(C·T^2·d + T·C^2·d)$|$O(C·T^2·d)$|$O(C·T^2·d)$|
>
> Where $C$ is the number of variables input to the backbone and $d$ is the size of the hidden dimension.
>
> To further demonstrate that STAR introduces only a marginal performance overhead, we summarize the number of parameters (M), training time per epoch (s), and inference time per sample (ms) for models using DADA and Moment as backbones on the SMAP and SWAT datasets in the table below. Note that rows labeled '(Full fine-tuning)' indicate that all parameters are fine-tuned, whereas the remaining configurations involve fine-tuning only the adapter parameters.
>
> |Dataset|SMAP|||SWaT|||
> |:-|:-|:-|:-|:-|:-|:-|
> |**Metric**|**Parameters**|**Training time**|**Inference time**|**Parameters**|**Training time**|**Inference time**|
> |**DADA** (Full fine-tuning)|1.84|18.51|9.26|1.84|91.90|19.63|
> |$~+$ **LoRA** (Fine-tuning)|1.91|8.61|10.01|1.95|71.23|19.94|
> |$~+$ **STAR** (Fine-tuning)|2.24|10.44|11.64|2.11|78.92|20.11|
> |**Moment** (Full fine-tuning)|35.34|40.66|8.06|35.34|227.43|35.53|
> |$~+$ **LoRA** (Fine-tuning)|35.53|19.72|11.07|35.57|169.41|37.67|
> |$~+$ **STAR** (Fine-tuning)|36.08|22.66|11.60|36.01|198.87|40.91|
>
> The results demonstrate that our method outperforms LoRA (Table 1, Lines 432–449) while maintaining comparable efficiency, achieving faster training speeds than full fine-tuning with only minimal inference overhead.
>
>
> We have also incorporated this efficiency analysis into our manuscript (Lines 500–514).
>
> ---
> ### Response to Q5: Visualizations
>
> **Differences in anomaly scores across states:**
> As shown in the original Figure 6 (Lines 949-964), the model integrated with STAR produces significantly different anomaly scores at time steps 40 and 70, despite the numerical variable values being similar at these timestamps. This demonstrates our method's ability to perceive distinct states and effectively integrate this contextual information into the reconstruction process of numerical variables, thereby influencing the final anomaly scores.
>
> **Differences in adapter activations across states:**
>
> As shown in the newly added Appendix B.6 (Lines 1062-1104), to analyze the bottleneck activations under different states,
> we visualize both the raw time series data and the corresponding mask vector.  This mask vector is conditioned on state variables and applied to matrix $\boldsymbol{R}$, which is inserted between matrices $\boldsymbol{A}$ and $\boldsymbol{B}$ in the Conditional Bottleneck Adapter to adaptively modulate the bottleneck, as defined in Eq. 9.
>
> Figure 11 illustrates the raw time series, highlighting the relationship between numerical and state variables in the Genesis dataset. When all state variables are zero, the numerical variable exhibits complex and rapid fluctuations; otherwise, it displays a stable pattern. We specifically selected representative timestamps $T_1$ and $T_3$ to exemplify the rapid fluctuation pattern, and $T_2$ and $T_4$ for the relatively stable pattern, as indicated by the purple markers in the figure.
>
> Figure 12 illustrates the mask vector of matrix $\boldsymbol{R}$ corresponding to timestamps $T_1$, $T_2$, $T_3$, and $T_4$, along with its masking ratio. This masking ratio reflects the bottleneck size of the weights in the Conditional Bottleneck Adapter.
>
> In instances with a rapid fluctuation pattern ($T_1$ and $T_3$), the numerical variable is relatively complex and requires more temporal information for reconstruction. Consequently, the corresponding masking ratio is lower, indicating a larger bottleneck size. In contrast, $T_2$ and $T_4$ represent the stable pattern, where the numerical variable is relatively simple and requires less temporal information. This results in a higher masking ratio, signifying a smaller bottleneck size.
>
> This phenomenon justifies our strategy of dynamically adjusting the bottleneck size according to the system state.
>
> **Thanks for your sincere suggestions again! If you have any additional questions, we can have further discussions!** 😊

---

### Author Response · Authors · 2025-12-02
**Rebuttal Summary (Pre-Incident Scores: 2, 4, 6, 8; Average 5.0)**

**Dear Reviewers, ACs, SACs, and PCs,**

We sincerely appreciate your dedication to reviewing our work and your valuable feedback.

We were sorry to learn about the recent technical issues with OpenReview, and we fully support the remedial actions proposed by the committee. To assist in the final assessment of our submission, we would like to summarize **the strengths of our work** and **the status of meaningful discussions** that concluded by **Nov. 26**.

**1. Strengths**

It is encouraging to see that reviewers acknowledged the practical value and novelty of our work, particularly highlighting:

* **Significant Practical Relevance**: The work addresses a critical yet overlooked gap in real-world industrial scenarios by distinguishing discrete state variables from numerical ones. (**cWBj, 3Yzz, taXc**)

* **Novel & Sound Methodology**: The proposed STAR framework is recognized as a well-motivated, plug-and-play solution. The design of STAR is considered novel and technically sound. (**taXc, 3Yzz**)

* **Effective Performance Improvements**: Extensive experiments on multiple real-world datasets consistently demonstrate that STAR  boosts the anomaly detection capabilities. (**cWBj, oaB6**)


**2. Score Improvement as of Nov. 26**

We summarize the score changes resulting from the discussion phase (up to **Nov. 26**) to reflect the true evaluation of the revised manuscript:

**Table 1: Score Summary**

| Reviewer | Initial Score | Score After Rebuttal | Discussion Status | Date of Update |
| :--- | :---: | :---: | :--- | :---: |
| **cWBj** | 6 | **8** $\textcolor{green}{⬆︎}$ | ✅Confirmed: Concerns Addressed  | **Nov. 26** |
| **taXc** | 6 | 6 | ✅Confirmed: Concerns Addressed  | **Nov. 26** |
| **3Yzz** | 4 | 4 |🔄 Detailed Response Provided (No Reply)   | - |
| **oaB6** | 2 | 2 |🔄 Detailed Response Provided (No Reply)| - |
| **Average** | 4.5 | **5.0** $\textcolor{green}{⬆︎}$ | | |

**3. Rebuttal Outcome & Discussion Status**

Positive Confirmation from Reviewers:

As of **Nov. 26**, we are pleased that **Reviewers cWBj (Score: 6 to 8) and taXc (Score: 6)** have explicitly confirmed that **their concerns were addressed**.

Regarding Reviewers with No Response:

Although **Reviewers oaB6 (Score: 2) and 3Yzz (Score: 4)** have not yet responded to our rebuttal, we have provided comprehensive responses to their specific concerns. Below we summarize our responses to each point raised:

**Table 2: Response to Reviewer oaB6 (Score: 2)**

| ID | Reviewer's Concern | Our Response Summary |
| :--- | :--- | :--- |
| **W1** | Comparison with previous works. | Provided **detailed explanations** and **two comparison tables** to demonstrate novelty. |
| **W2,&nbsp;Q3,4** | Unclear description of a few technical details. | Provided **detailed clarifications** on these technical details. |
| **W3,&nbsp;Q1,2** | Writing typos. | Thoroughly **revised** our manuscript. |
| **W4** | Runtime and complexity analysis. | Provided **runtime comparisons** and **theoretical complexity analysis** to demonstrate efficiency. |
| **Q5** | More visualizations. | Added **Appendix B.6** with **further visualizations**. |


**Table 3: Response to Reviewer 3Yzz (Score: 4)**

| ID | Reviewer's Concern | Our Response Summary |
| :--- | :--- | :--- |
| **W1** | Computational cost. | Provided **runtime comparisons** and **theoretical complexity analysis** to demonstrate efficiency. |
| **W2** | 1.Motivation & Real-world Justification.| Clarified the motivation with **real-world empirical evidence (Fig. 1c).** |
| | 2. Marginal performance gains in a couple of cases. | Demonstrated ability to capture **additional anomaly types** and reduce false alarms in these cases (visualized in **Appendix B.5**). |
| **W3** | Additional experiments. | Provided the requested **sensitivity analysis** for $\lambda_1$ and $\lambda_2$. |
| **Q1** | Distinguishing Numeral and State variables. | We analyzed the **value ranges** to demonstrate that Numeral and State variables are easily distinguishable (Tab. 6 Lines 787-797). |




We once again thank all reviewers for their valuable comments and the ACs for navigating this situation. We look forward to a further assessment of our submission.

Best regards,

Authors of STAR

---

### Meta-Review · Area_Chair_4DTi · 2026-01-04

**Summary:**

There are various concerns about computational cost, comparisons with baselines, unclear technical details, and various experimental result clarifications. The authors did a solid effort to address these concerns and indeed a reviewer increased their score. The more negative reviewers did not participate in the discussion, unfortunately, and it's unclear if the responses would have addressed comments and result in score increases. Even if we assume one would increase their score, the overall score would be in borderline/low acceptance category and hence very uncertain it would be accepted in the end. I hope the authors would address the comments and find a new venue for their work.

**Reviewer Concerns:**

The major concern has to do with technical novelty and it's unclear if the reviewer who is most negative would have raised their score

**Reviewer Scores:**

One reviewer raised their score. However, the most negative reviewer did not participate in the discussion and even under the assumption of increased score, the overall score would still be in borderline category.

---

### Decision · Program_Chairs · 2026-01-26

Reject